# Large protein organelles form a new iron sequestration system with high storage capacity

Tobias W Giessen[1,2,3]\*, Benjamin J Orlando[4], Andrew A Verdegaal[1,2†], Melissa G Chambers[4‡], Jules Gardener[5], David C Bell[5,6], Gabriel Birrane[7], Maofu Liao[2,4]\*, Pamela A Silver[1]\*

[1]Department of Systems Biology, Harvard Medical School, Boston, United States; [2]Wyss Institute for Biologically Inspired Engineering, Harvard University, Boston, United States; [3]Department of Biomedical Engineering, University of Michigan, Ann Arbor, United States; [4]Department of Cell Biology, Harvard Medical School, Boston, United States; [5]Center for Nanoscale Systems, Harvard University, Cambridge, United States; [6]School of Engineering and Applied Sciences, Harvard University, Cambridge, United States; [7]Department of Medicine, Beth Israel Deaconess Medical Center, Harvard Medical School, Boston, United States

\*For correspondence:
tgiessen@umich.edu (TWG);
maofu_liao@hms.harvard.edu
(ML);
pamela_silver@hms.harvard.edu
(PAS)

Present address: †Program in
Biological and Biomedical
Sciences, Yale University, New
Haven, United States; ‡Center
for Structural Biology, Vanderbilt
University, Nashville, United
States

Competing interests: The
authors declare that no
competing interests exist.

Reviewing editor: Werner
Kühlbrandt, Max Planck Institute
of Biophysics, Germany

**Abstract** Iron storage proteins are essential for cellular iron homeostasis and redox balance. Ferritin proteins are the major storage units for bioavailable forms of iron. Some organisms lack ferritins, and it is not known how they store iron. Encapsulins, a class of protein-based organelles, have recently been implicated in microbial iron and redox metabolism. Here, we report the structural and mechanistic characterization of a 42 nm two-component encapsulin-based iron storage compartment from *Quasibacillus thermotolerans*. Using cryo-electron microscopy and x-ray crystallography, we reveal the assembly principles of a thermostable T = 4 shell topology and its catalytic ferroxidase cargo and show interactions underlying cargo-shell co-assembly. This compartment has an exceptionally large iron storage capacity storing over 23,000 iron atoms. Our results reveal a new approach for survival in diverse habitats with limited or fluctuating iron availability via an iron storage system able to store 10 to 20 times more iron than ferritin.
DOI: https://doi.org/10.7554/eLife.46070.001

## Introduction

Iron is essential to virtually all organisms on earth. It is needed for a wide variety of catalytic and redox processes ranging from cellular energy production via oxidative phosphorylation to oxygen transport by hemoglobin (*Sánchez et al., 2017*). However, the same properties that make iron useful for cellular metabolism can result in toxicity under aerobic conditions (*Sánchez et al., 2017*). Ferrous iron ($Fe^{2+}$) is easily oxidized to insoluble ferric iron ($Fe^{3+}$) resulting in the formation of harmful precipitates and reactive oxygen species (ROS) via Fenton chemistry (*Dixon and Stockwell, 2014*). Cells have evolved to cope with these problems by strictly controlling the intracellular concentration and reactivity of free iron (*Crichton, 2002*). Ferritin proteins are used as the main iron storage system by animals, plants and most microbes (*Arosio et al., 2017*). The main ferritin-like proteins involved in iron storage are ferritin (Ftn), bacterioferritin (Bfr) and DNA-binding proteins from starved cells (Dps) all able to oxidize $Fe^{2+}$ to $Fe^{3+}$ via a ferroxidase activity (*Andrews, 2010*). While Ftn and Bfr are primarily used as a dynamic iron storage (*Honarmand Ebrahimi et al., 2015*), the main function of Dps proteins is to counteract oxidative stress (*Chiancone et al., 2004*). Ferritins (Ftn and Bfr) assemble into 24 subunit protein compartments up to 12 nm in diameter able to store 2000 to 4,000 Fe atoms

**eLife digest** People often think of the cell as the basic unit of life. Despite this, individual cells are also subdivided into many compartments, called 'organelles' because they act like the internal organs of the cell. For example, organelles can break down nutrients, store information in the form of DNA, or help remove waste. Even bacterial cells, despite being smaller and simpler than most other cell types, contain organelle-like structures. These are tiny compartments, termed protein organelles, which are enclosed by 'shells' made from self-assembling proteins within the cell.

Cells need iron to carry out the chemical reactions necessary for life. Iron is therefore an essential nutrient, but it can also be toxic if not stored properly inside the cell. Cells often solve this problem by locking iron away inside small, specialised protein cages called ferritins until it can be used. Most organisms, from humans to bacteria, have ferritins, but some do not, and the way these organisms store iron remains largely unknown.

The bacterium *Quasibacillus thermotolerans* is an example of an organism that lacks ferritins. However, it does contain a recently discovered type of protein organelle, called an encapsulin. Giessen et al. wanted to find out more about the structure of this protein organelle, and to determine if it helped these bacteria store iron.

*Q. thermotolerans*' encapsulin turned out to be the largest of its kind discovered to date. Detailed imaging experiments, using a combination of electron microscopy and X-ray- based techniques, revealed that the protein shell of the encapsulin had an overall structure resembling chain mail and contained multiple pores. These pores were negatively charged, meaning that they could efficiently attract iron (which has a positive charge) and funnel it into the interior of the compartment. The compartment itself was able to store at least 20 times more iron than ferritins, making this encapsulin one of the most efficient methods of iron storage in any cell.

These findings will help us better understand how bacteria that lack ferritins cope with the problem of iron storage. In the future, encapsulins could also be used as a target for new therapies to fight bacterial infections, or even as the building blocks for microscopic chemical reactors or 'storage facilities' in industrial applications.

DOI: https://doi.org/10.7554/eLife.46070.002

in their interior (*Andrews, 1998*; *Harrison and Arosio, 1996*). However, some organisms do not encode ferritin genes in their genomes and their iron storage systems have remained elusive.

A newly discovered class of protein organelles called encapsulin nanocompartments have been shown to be involved in microbial iron storage and redox metabolism (*Giessen and Silver, 2017*; *He et al., 2016*; *McHugh et al., 2014*; *Sutter et al., 2008*). Previously reported encapsulins share an HK97 phage-like fold and self-assemble from a single capsid protein into icosahedral compartments between 24 and 32 nm in diameter with triangulation numbers of T = 1 (60 subunits) and T = 3 (180 subunits), respectively (*Akita et al., 2007*; *McHugh et al., 2014*; *Sutter et al., 2008*). Their key feature is the ability to specifically encapsulate cargo proteins (*Figure 1a*). Encapsulation is mediated by short C-terminal sequences referred to as targeting peptides (TPs) (*Sutter et al., 2008*; *Tamura et al., 2015*). Genes encoding encapsulin shell proteins and dedicated cargo proteins are organized in co-regulated operons (*Giessen and Silver, 2017*; *Sutter et al., 2008*). So far, operons involved in hydrogen peroxide and nitric oxide detoxification as well as iron mineralization have been reported (*Nichols et al., 2017*). The main cargo protein-types described to date are DyP-type peroxidases, hemerythrins and different classes of ferritin-like proteins (*Contreras et al., 2014*; *Giessen and Silver, 2017*; *McHugh et al., 2014*; *Rahmanpour and Bugg, 2013*). We have identified a novel type of encapsulin operon involved in iron metabolism in a range of Firmicutes we term the Iron-Mineralizing Encapsulin-Associated Firmicute (IMEF)-system (*Giessen and Silver, 2017*).

Here, we report the structural and mechanistic characterization of the IMEF-system found in *Quasibacillus thermotolerans* (*Qs*), an organism that does not encode any ferritins in its genome. We show that this encapsulin-based system self-assembles into a thermostable 42 nm 9.6 MDa protein compartment with a novel T = 4 topology able to mineralize and store an exceptionally large quantity of iron.

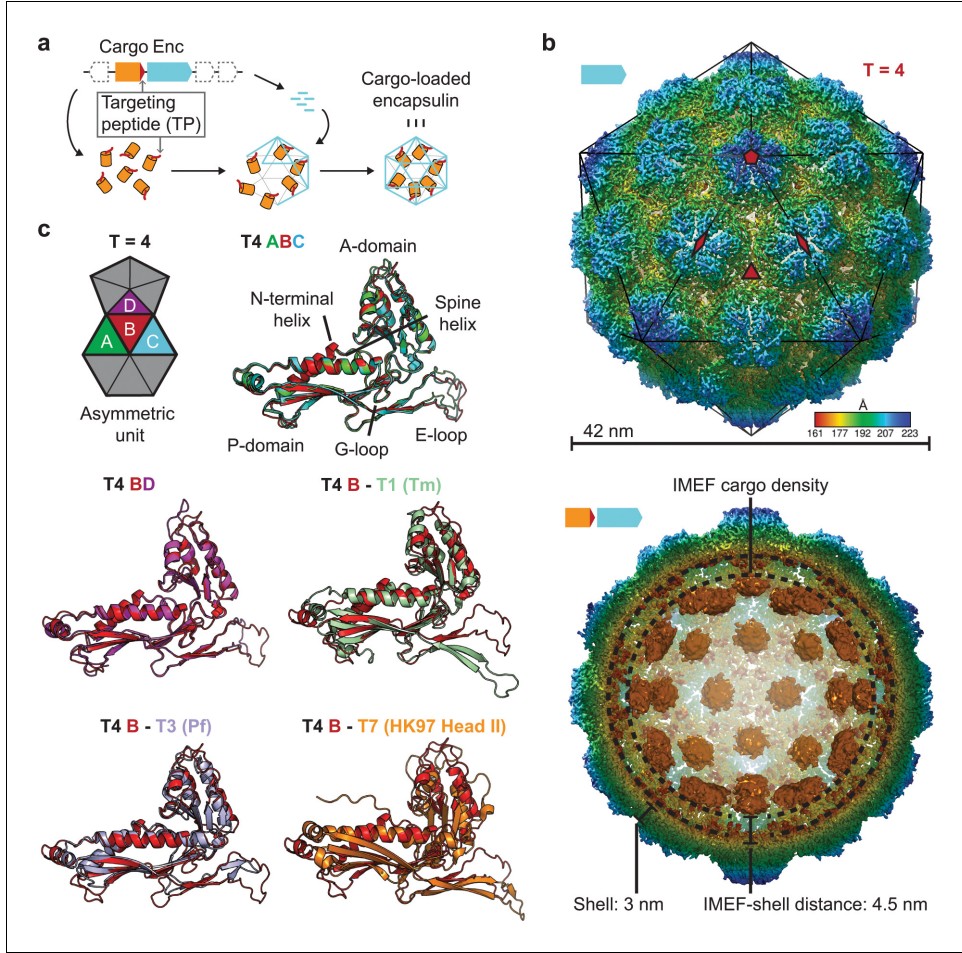

**Figure 1.** Overall architecture of the cargo-loaded T = 4 encapsulin. (**a**) Schematic diagram of a core encapsulin operon and targeting peptide (TP)-dependent cargo encapsulation. (**b**) Surface view of the cryo-EM map of the *Qs* T = 4 encapsulin shell (top) and inside view of cargo-loaded encapsulin (bottom). 5-, 3- and 2-fold symmetry axes are indicated by red symbols. The overall icosahedral symmetry is highlighted by black lines representing icosahedral facets. Cargo-densities are shown in orange while the shell is radially colored. To depict the complete cargo-loaded compartment, a 10 Å filtered map highlighting the cargo was combined with the 3.85 Å map of the shell. (**c**) Asymmetric unit of the T4 encapsulin shell and structural alignment of the four unique T4 shell monomers with one another and with the *T. maritima* (Tm) T = 1 monomer (3DKT), the *P. furiosus* (Pf) T = 3 monomer (2E0Z) and the HK97 bacteriophage Head II T = 7 monomer (2FT1).

DOI: https://doi.org/10.7554/eLife.46070.003

The following figure supplements are available for figure 1:

**Figure supplement 1.** Identification of IMEF operons and confirmation of protein compartment formation.
DOI: https://doi.org/10.7554/eLife.46070.004

**Figure supplement 2.** Supplementary cryo-EM data.
DOI: https://doi.org/10.7554/eLife.46070.005

**Figure supplement 3.** Symmetry expansion classification of cargo density.
DOI: https://doi.org/10.7554/eLife.46070.006

**Figure supplement 4.** Local resolution maps of the cargo-loaded T = 4 IMEF encapsulin.
DOI: https://doi.org/10.7554/eLife.46070.007

## Results and discussion

### Discovery and computational analysis of IMEF operons

IMEF-systems are found in Firmicute genomes and their operon organization indicates a function in dynamic iron storage. To investigate the distribution of IMEF-systems in microbes, we carried out

BLASTp searches using IMEF cargo proteins as queries and identified 71 operons in a range of Firmicutes including *Qs* (*Figure 1—figure supplement 1a*). The core operon consists of the encapsulin capsid protein and the IMEF cargo protein with 70% of operons also encoding a 2Fe-2S ferredoxin homologous to bacterioferritin-associated ferredoxins (Bfd). Bfd proteins are involved in the mobilization of iron under iron-limited conditions (*Yao et al., 2012*). In addition, 31% of operons are associated with proteins similar to ferrochelatases involved in catalyzing the insertion of ferrous iron into protoporphyrins (*Dailey et al., 2000*). The majority of IMEF-encoding genomes do not contain any ferritin or bacterioferritin genes (*Supplementary file 1*). Most IMEF genomes do however contain Dps-encoding genes. Overall, the operon organization of IMEF-systems and the lack of other known primary iron storage proteins indicate a function for IMEF-systems in dynamic iron storage similar to that of Ftn and Bfr.

## Overall structure of the cargo-loaded IMEF encapsulin

Using a recombinant system for the expression of the two-gene IMEF operon containing the IMEF cargo protein gene and the encapsulin capsid protein gene, we produced homogeneous IMEF cargo-loaded encapsulins (*Figure 1—figure supplement 1b*). Through single-particle cryo-EM analysis, we determined the structure of the *Qs* IMEF encapsulin shell at an overall resolution of 3.85 Å (*Figure 1—figure supplement 2a* and *Supplementary file 2*). The IMEF encapsulin self-assembles into a 240-subunit icosahedral compartment with a diameter of 42 nm (*Figure 1b* and *Figure 1—figure supplement 2a–d*). The IMEF compartment is substantially larger than previously reported encapsulins and possesses a triangulation number of T = 4 instead of T = 1 (60 subunits, 24 nm) or T = 3 (180 subunits, 32 nm) and represents the largest encapsulin compartment reported to date (*Figure 1—figure supplement 2e*). The shell is composed of 12 pentameric and 30 hexameric capsomers occupying icosahedral vertices and faces, respectively. In contrast, T = 1 encapsulins consist of only 12 pentameric capsomers while the T = 3 encapsulin shell is made up of 12 pentameric and 20 hexameric capsomers. The T = 4 IMEF-system consequently possesses an internal volume 530% and 220% larger than that of T = 1 and T = 3 encapsulins, respectively. The 5-fold symmetry axes are located at the pentameric vertices while 3-fold symmetry axes are present at all interfaces where three hexameric capsomers meet. The center of each hexameric capsomer corresponds to an icosahedral edge possessing 2-fold symmetry. The icosahedral asymmetric unit consists of one pentameric and three hexameric monomers (*Figure 1b* and *Figure 1—figure supplement 2c*). Symmetrically arranged lower resolution density (ca. 10 Å) representing the IMEF cargo is visible in the compartment interior (*Figure 1b* and *Figure 1—figure supplement 2d*). 42 distinct densities, one for each capsomer of the T = 4 structure, can be observed. No connection of cargo and shell density is visible, likely due to averaging or the flexibility of a 37 amino acid linker preceding the IMEF targeting peptide that directs and anchors the IMEF cargo to the shell interior. Averaging and linker flexibility likely also contribute to the lower resolution observed for the interior IMEF densities. The distance between the shell and cargo densities is 4.5 nm which can be bridged by the 37 amino acid linker. To further investigate and better resolve the cargo densities, we applied an approach combining symmetry expansion and focused classification with residual signal subtraction (*Figure 1—figure supplement 3*). This approach was able to separate cargo densities bound at slightly different locations indicating that the symmetry observed for the cargo densities (*Figure 1b*) is a result of averaging. The observed non-symmetrical densities are still weak compared to the shell density. At low threshold values possible connections between cargo densities and the shell are visible, potentially representing the linker connecting the cargo with the bound TP (*Figure 1—figure supplement 3*).

The four capsid proteins of the asymmetric unit adopt different conformations with significant differences found in the E-loop and A-domain (*Figure 1c*). E-loops are located at capsomer interfaces and their relative orientation plays a key role in determining the overall topology and triangulation number of encapsulin compartments as evidenced by comparison of the IMEF T = 4 monomer with T = 1 (*Thermotoga maritima*), T = 3 (*Pyrococcus furiosus*) and T = 7 (HK97 phage) capsid proteins (*Figure 1c*). A-domain loops form compartment pores and are likely adapted to optimize the particular function of a given encapsulin, for example ROS detoxification or iron mineralization. In addition, local resolution maps indicate that E-loops and A-domain loops represent the most flexible parts of the shell which suggests a certain structural flexibility of the pores formed by A-domain loops (*Figure 1—figure supplement 4*).

## Pores in the IMEF encapsulin shell

The IMEF encapsulin shell contains negatively charged pores at the 3- and 5-fold symmetry axes. The surface view of the intact shell (*Figure 2—figure supplement 1a*) shows a tight packing with pores at the 3- and 5-fold symmetry axes and at the interface between two hexameric and one pentameric capsomer (pseudo 3-fold) representing the only conduits to the interior. Similarly, pores at the symmetry axes were also reported for T = 1 and T = 3 encapsulin systems. All pores in the IMEF-system are negatively charged on both the exterior and interior surface due to the presence of conserved aspartate, glutamate and asparagine residues (*Figure 2a,b*, *Figure 2—figure supplement 1b* and *Figure 2—figure supplement 2*). This is similar to the negatively charged pores in ferritin systems that guide positively charged iron to the ferritin interior (*Arosio et al., 2017*). In no other encapsulin system are all pores negatively charged indicating that pores in the IMEF-system are optimized for attracting and channeling positively charged ions. The 2-fold pores observed at the interface of two capsomers in T = 1 and T = 3 encapsulins are not present in the IMEF-system (*Nichols et al., 2017*). The 3-fold pore forms the largest channel to the IMEF compartment interior and is 7.2 Å wide at its narrowest point, substantially larger than previously reported encapsulin pores. Extra cryo-EM density is observed at the center of both the 3-fold and 5-fold pores. This could be a result of averaging accentuating noise on symmetry axes or potentially represent bound ions (e.g. $Fe^{2+/3+}$) or even water molecules. The 2-fold symmetry axes at the center of hexameric capsomers also represent potential channels, as observed in T = 3 systems (*Nichols et al., 2017*), but the conformation of two asparagine side chains prevents the formation of a 2-fold opening in the T = 4 shell leading to a closed pore (*Figure 2c*). This observation combined with the flexibility observed for loops around the 2- and 5-fold symmetry axes in local resolution maps (*Figure 1—*

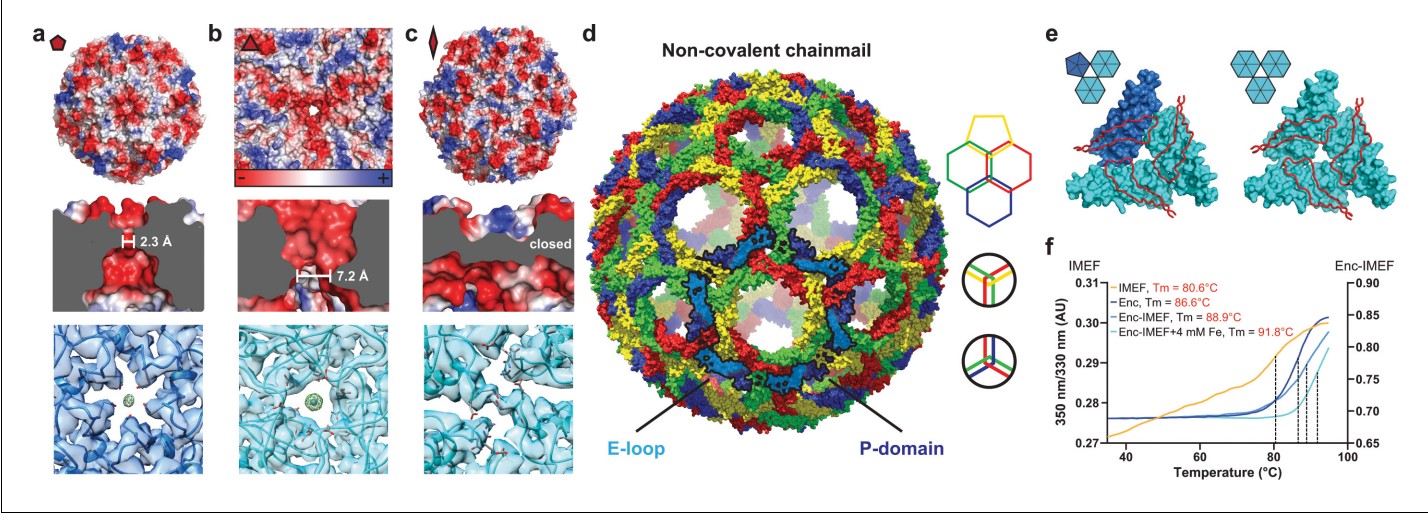

**Figure 2.** Non-covalent chainmail topology, thermal stability and pores of the T = 4 encapsulin shell. (a, b and c) Electrostatic surface representation of the 5-fold (d) and 3-fold (e) T = 4 shell pores and the 2-fold symmetry axis (f). Outside views showing negatively charged pores (top) with no pore opening observed at the two-fold symmetry axis, cutaway side view highlighting the narrowest point of the pores (middle) and cryo-EM maps with fitted monomer models in ribbon representation (bottom). Additional cryo-EM density is observed at the center of both pores in interaction distance with the side chains of pore residues (5-fold: Asn200, 3-fold: Asp9, Asp71, Glu251 and Glu252, shown in stick representation). (d) Chainmail network mediated by E-loop and P-domain interactions. Only E-loops and P-domains are shown. E-loops and P-domains of the outlined ring belonging to the same monomer are located next to one another and are shown in light and dark blue, respectively. (e) Extended E-loop interactions interlock neighboring capsid monomers at the two unique three-fold interfaces. Each E-loop interacts with two P-domains. (f) Representative thermal unfolding curves for *Qs* T = 4 encapsulin components determined via differential scanning fluorimetry. Tm: midpoint of the thermal unfolding curve.

DOI: https://doi.org/10.7554/eLife.46070.008

The following figure supplements are available for figure 2:

**Figure supplement 1.** Structural details of the T4 encapsulin shell.

DOI: https://doi.org/10.7554/eLife.46070.009

**Figure supplement 2.** Sequence alignment of representative T = 1, T = 3 and T = 4 encapsulin capsid proteins.

DOI: https://doi.org/10.7554/eLife.46070.010

*figure supplement 4*) could indicate the presence of gated pores in encapsulins that may regulate ion flux to the compartment interior, similar to some ferritins (*Theil et al., 2008*).

## Non-covalent chainmail and thermal stability of the IMEF-system

The IMEF compartment possesses a non-covalent chainmail topology and is highly thermostable. E-loops and P-domains of neighboring capsid monomers arrange head to tail to form interlocking concatenated rings resulting in a non-covalent chainmail topology (*Figure 2d*) (*Zhang et al., 2013*). This architecture has only been observed in a number of viral capsids including the HK97 bacteriophage but not in a bacterial system. In contrast to HK97 where an isopeptide bond covalently links E-loops and P-domains (*Duda, 1998*), the IMEF encapsulin uses non-covalent interactions. At each 3-fold pore, E-loops connect with two neighboring P-domains including the G-loop conserved in T = 4 encapsulins and their interfaces contain complementary electrostatic as well as aromatic and potential anion-π interactions (*Figure 2e* and *Figure 2—figure supplement 1c,d*) (*Philip et al., 2011*). The IMEF cargo protein shows a linear unfolding curve starting at ca. 40°C and extending to ca. 75°C followed by a hyperbolic increase leading to a midpoint of the thermal unfolding curve of 80.6°C. The shell protein is highly thermostable with a melting temperature of 86.6°C, respectively (*Figure 2f*). A stabilizing effect is observed for the cargo-loaded compartment (88.9°C). Compartments isolated from high iron conditions show even greater thermal stability (91.8°C) likely due to the internal cavity being stabilized by mineralized material.

## Structure and analysis of the IMEF cargo protein

Sequence and x-ray structure analysis show that the IMEF cargo represents a distinct class of ferritin-like protein (Flp) with an unusual ferroxidase center. Phylogenetic analysis revealed that the IMEF cargo protein is a member of the Flp superfamily and is most closely related to Dps proteins (*Figure 3a* and *Supplementary file 3*) but no known ferroxidase motifs could be detected based on the primary sequence alone (*Andrews, 2010*). IMEF proteins form a separate clade distinct from other Flp proteins associated with encapsulin systems. All IMEF proteins share a conserved C-terminal TP (*Figure 3b*). We determined the x-ray crystal structure of the IMEF cargo to a final resolution of 1.72 Å (*Figure 3c* and *Supplementary file 4*). The cargo adopts a four-helix bundle fold characteristic of other members of the Flp superfamily and forms a dimer with two Fe atoms bound at the subunit interface creating a ferroxidase site based on an alternative ferroxidase sequence motif (*Figure 3d*, *Figure 3—figure supplement 1a,b*). This leads to a combined molecular weight of the fully cargo-loaded IMEF compartment of 9.6 MDa (42 × cargo dimer [22.6 kDa]+240 × capsid protein, [32.2 kDa]). Through structure and sequence analysis, we identified a set of conserved residues involved in the formation of the dinuclear ferroxidase center. This IMEF ferroxidase motif differs from known examples and represents an alternative way of forming an inter-subunit ferroxidase center (*Figure 3d*). Due to flexibility, the C-terminal linker and TP are not resolved in the cargo x-ray structure in accordance with observations from our cryo-EM analysis. Removal of the 13 C-terminal residues results in empty encapsulin shells confirming that the IMEF TP is necessary for cargo encapsulation (*Figure 3e*).

## TP-mediated cargo-shell co-assembly

Additional cryo-EM density around the 2- and 5-fold symmetry axes reveals TP-binding sites and illuminates cargo-shell co-assembly. Through analysis of the T = 4 cryo-EM map, additional densities were identified that could not be explained by the encapsulin capsid protein (*Figure 3f*). These densities represent bound TPs anchoring IMEF cargo to the interior surface of the compartment. Even though only 42 cargo densities are observed, TP densities can be found at all 240 capsid monomers indicating averaging during cryo-EM reconstruction. Strong TP density is observed for all 180 monomers that are part of 2-fold symmetrical hexameric capsomers (*Figure 3f*) while substantially weaker density is found for TPs bound to the 60 pentameric monomers (*Figure 3—figure supplement 1c*) thus revealing higher occupancy and preferential TP binding around 2-fold symmetry axes which can be explained by different binding site conformations (*Figure 3—figure supplement 1c–e*) and higher local shell mobility (*Figure 1—figure supplement 4*). The main TP binding sites surrounding the 2-fold symmetry axes are formed by conserved residues of the P-domain and N-terminal helix (*Figure 2—figure supplement 2*) similar to the *T. maritima* T = 1 encapsulin system (*Sutter et al.,*

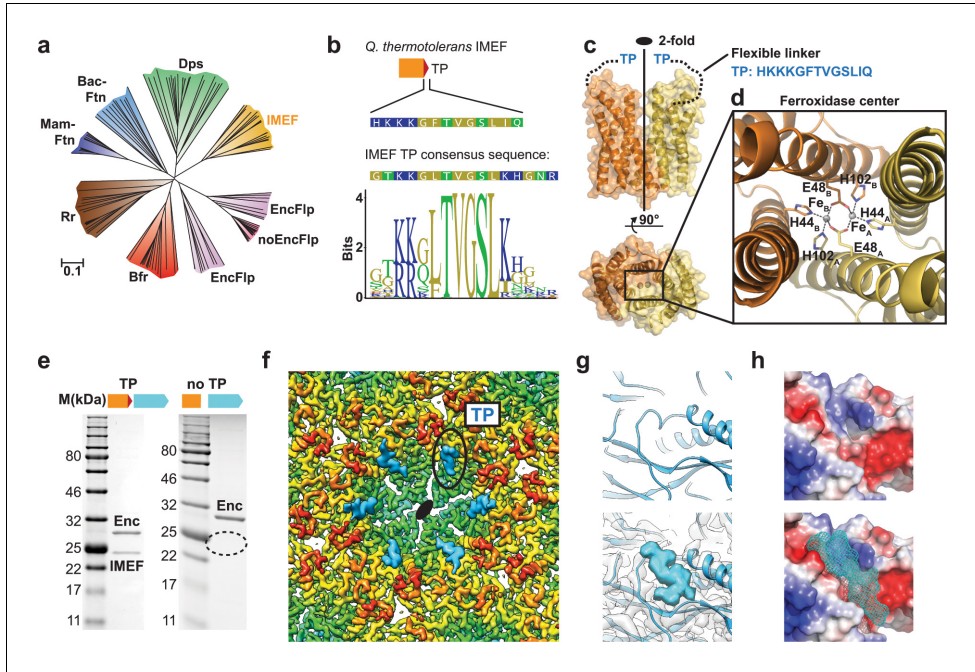

**Figure 3.** Structure and analysis of the IMEF cargo protein and TP-mediated cargo-shell co-assembly. (**a**) Neighbor-joining phylogeny (cladogram) of protein classes involved in iron metabolism that are part of the Flp superfamily. Scale bar: amino acid substitutions per site. EncFlp: Flps found within encapsulin operons containing TPs, noEncFlp: Flps found outside encapsulin operons not containing TPs, Rr: rubrerythrins, Mam-Ftn: mammalian ferritins, Bac-Ftn: bacterial ferritins. (**b**) TP sequence of the *Qs* IMEF cargo protein and TP sequence logo highlighting strong sequence conservation. (**c**) X-ray crystal structure of the *Qs* IMEF cargo. (**d**) Di-iron ferroxidase active site of the IMEF cargo. The iron-coordinating residues are shown in stick representation. (**e**) SDS-PAGE gels of purified encapsulins showing that co-purification is dependent on the presence of the TP. (**f**) Cryo-EM map interior view of the 2-fold symmetry axis with TP density shown in cyan. (**g**) Close-up of additional cryo-EM density observed around the 2-fold symmetry axis. (**h**) Electrostatic surface representation of the TP binding site without (top) and with (bottom) TP. The 7 C-terminal IMEF residues are shown as a surface mesh.
DOI: https://doi.org/10.7554/eLife.46070.011

The following figure supplement is available for figure 3:

**Figure supplement 1.** Biochemical and structural analysis of IMEF cargo loading.
DOI: https://doi.org/10.7554/eLife.46070.012

*2008*). No TP binding site has been identified for T = 3 encapsulins yet. The presence of the N-terminal helix and the resulting binding site seems to generally underpin encapsulins' ability to interact with TPs and encapsulate cargo proteins. The TP residues TVGSLIQ were tentatively built and refined into the additional density present at hexameric capsomers producing a model with good geometry (*Figure 3—figure supplement 1e*). The TP binds to a surface groove based on shape complementarity and two key ionic interactions with highly conserved positively charged residues locking the TP in place.

## Iron mineralization and storage by the IMEF-system

Heterologous expression of the IMEF core operon in *E. coli* leads to in vivo formation of large Fe- and P-rich electron-dense particles. Thin section negative stain transmission electron microscopy (TEM) of *E. coli* cells grown in Fe-rich (4 mM) medium and expressing the *Qs* IMEF core operon results in the formation of clusters of large intracellular electron-dense particles (*Figure 4a* and *Figure 4—figure supplement 1a*). Scanning TEM-energy-dispersive x-ray spectroscopy (EDS) revealed that these particles primarily contain uniformly distributed Fe, P and O with an estimated Fe:P ratio near 1 (*Figure 4b*). Selected area electron diffraction (SAED) further indicates that this mineralized material is amorphous (*Figure 4—figure supplement 1b,c*), similar to bacterioferritin systems (*Andrews et al., 1993*). The high P content and amorphous cores described for the IMEF encapsulin

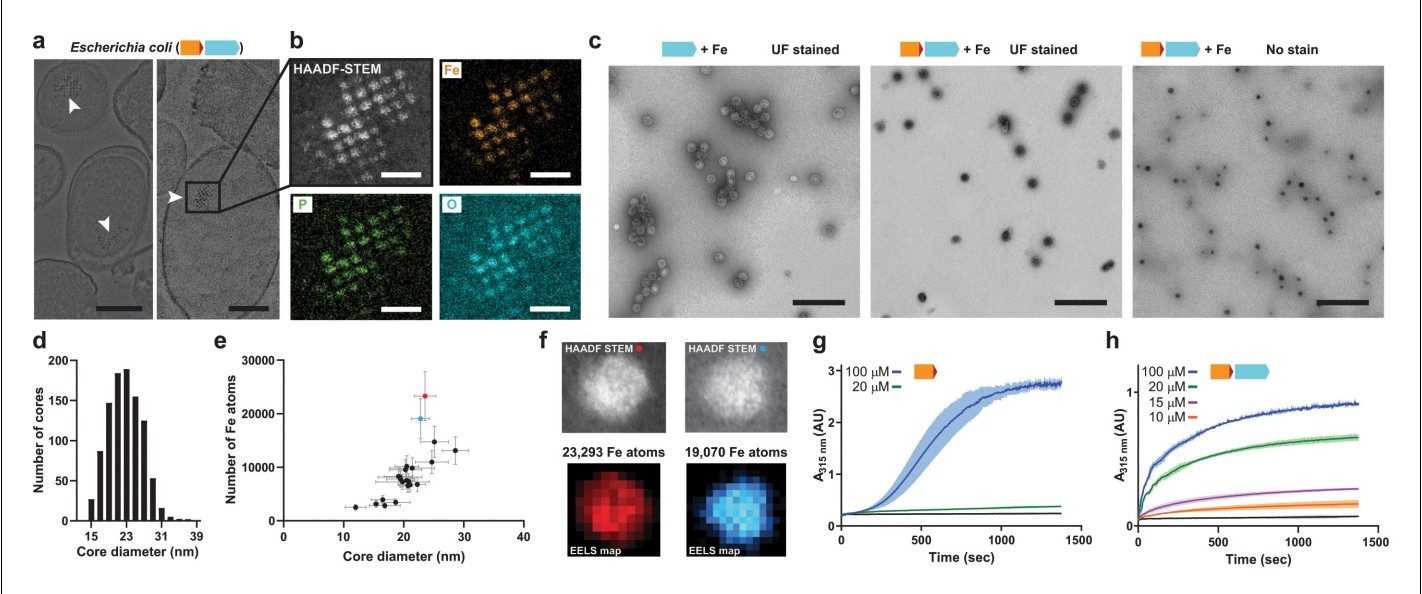

**Figure 4.** Mineralization of large iron-rich particles by the T = 4 encapsulin. (**a**) Thin section micrographs of *E. coli* heterologously expressing the *Qs* IMEF core operon. Electron-dense particles often cluster together in regular arrays. Scale bars: 500 nm (left), 400 nm (right). (**b**) Close-up high angle angular dark field (HAADF) scanning TEM and EDS maps of a cluster of particles showing Fe, P and O as the main particle constituents. Scale bars: 100 nm. (**c**) Micrographs of uranyl formate (UF)-stained encapsulins produced in and isolated from *E. coli* grown in high iron media expressing the capsid protein alone (left) or the core operon (middle and right). Without UF stain, electron-dense particles are clearly visible (right). Scale bars: 250 nm. (**d**) Size distribution of electron-dense particles in unstained micrographs. (**e**) Electron energy loss spectroscopy (EELS) of 22 select cores carried out on isolated encapsulin particles. (**f**) HAADF-STEM micrographs and EELS maps of the two highlighted cores from (**E**). (**g**) In vitro ferroxidase assay of purified IMEF cargo at different $Fe^{2+}$ concentrations. Mean values resulting from technical triplicates and error bands using standard deviation are shown. (**h**) Ferroxidase assay of cargo-loaded T = 4 encapsulin at different $Fe^{2+}$ concentrations.

DOI: https://doi.org/10.7554/eLife.46070.013

The following figure supplements are available for figure 4:

**Figure supplement 1.** Heterologous IMEF cargo-dependent *in vivo* minearlization and characterization of iron-rich particles.
DOI: https://doi.org/10.7554/eLife.46070.014

**Figure supplement 2.** Purification of iron-loaded T4 encapsulins from *E. coli* and identification of electron-dense particles in *Geobacillus*.
DOI: https://doi.org/10.7554/eLife.46070.015

**Figure supplement 3.** EDS and EELS analysis of purified iron-loaded T4 encapsulins.
DOI: https://doi.org/10.7554/eLife.46070.016

**Figure supplement 4.** Model explaining the observed ferroxidase activities of free IMEF cargo (left) and the cargo-loaded IMEF encapsulin (right).
DOI: https://doi.org/10.7554/eLife.46070.017

are similar to bacterioferritin systems (*Aitken-Rogers et al., 2004*; *Mann et al., 1986*). It has been hypothesized that amorphous material can be more readily mobilized under iron-limited condition than crystallized iron mineral (*Watt et al., 1992*; *Watt et al., 2010*).

The IMEF encapsulin mineralizes up to 30 nm Fe-rich cores in its interior with up to 23,000 Fe atoms stored per particle. IMEF encapsulins purified from *E. coli* grown under high Fe conditions contain electron dense cores visible in unstained samples with an average diameter of 23 nm (*Figure 4c,d* and *Figure 4—figure supplement 2a*). The largest observed particles are up to 30 nm in diameter. The theoretical size limit imposed by the T = 4 encapsulin protein shell is 36 nm and particles close to this size are observed in thin-sections of *Geobacillus* natively encoding the IMEF-system (*Figure 4—figure supplement 2b–d*). EDS analysis of particles isolated from *E. coli* and comparison with standards indicate a very similar elemental composition and elemental distribution as observed for thin section samples with a Fe:P ratio of 1:1.1 (*Figure 4—figure supplement 3a*). To determine the number of iron atoms stored per particle, we carried out electron energy loss spectroscopy (EELS) on purified Fe-loaded compartments (*Figure 4e,f* and *Figure 4—figure supplement 3b,c*). The highest observed number of stored Fe per particle was 23,293 (23.6 nm)

(*Supplementary file 5*). Extrapolating to the maximum theoretical particle diameter of 36 nm and the highest density observed (3.40 Fe atoms/nm$^3$) leads to a maximum number of Fe atoms that can be stored by the IMEF-system of around 83,000 (*Supplementary file 5*). Thus, IMEF-systems are able to store substantially more iron than any known ferritin system (2,000–4,000 Fe atoms) (*Andrews, 1998*; *Harrison and Arosio, 1996*).

To learn more about the mechanism of iron mineralization, we assayed peroxidase and ferroxidase activity. Due to the IMEF cargo being most closely related to Dps proteins we initially performed peroxidase assays using hydrogen peroxide as the oxidant. However, no peroxidase activity could be observed. Next, we assayed ferroxidase activity using $O_2$ as the oxidant. For the IMEF cargo alone, a sigmoidal ferroxidase iron oxidation curve was observed indicative of autocatalytic Fe oxidation taking place at newly formed mineral surfaces (*Bou-Abdallah et al., 2005*; *Sun and Chasteen, 1992*). However, assaying the IMEF cargo-loaded encapsulin results in a typical hyperbolic enzyme catalysis curve. These observations imply that the encapsulin shell controls the flux of iron to the inside of the compartment leading to a controlled and low concentration of soluble iron in the encapsulin interior. Therefore, the IMEF cargo protein is able to enzymatically oxidize the majority of ferrous iron before uncontrolled autocatalytic mineralization can lead to bulk precipitation of iron which would likely destroy the iron storage function of the IMEF-system (*Figure 4—figure supplement 4*).

Our structural model and functional analysis of the IMEF encapsulin system reveal an alternative way to store large amounts of Fe independent of ferritins. The IMEF-system can in principle store more than 20 times more Fe than Ftn or Bfr systems. In contrast to ferritin systems, IMEF encapsulins are two-component systems with the catalytic activity separated from the protein shell. The IMEF cargo protein is flexibly tethered and primarily localizes 4.5 nm away from the capsid interior. This suggests that once iron enters the encapsulin interior via pores, it diffuses to the ferroxidase active site of the IMEF cargo, making it necessary to strictly control interior iron concentration to prevent runaway mineralization. This is different compared with ferritin systems where the ferroxidase activity is part of the shell and negatively charged surface patches guide iron from the pores to ferroxidase sites.

It is striking that IMEF-systems are confined to spore-forming Firmicutes. They inhabit a broad range of habitats with many of them initially isolated from hot springs or soil, environments with often limited or fluctuating iron availability (*Colombo et al., 2014*; *Hou et al., 2013*; *Huang et al., 2013*). The ability to store a much larger amount of iron than other microbes might benefit IMEF-encoding organisms in these environments and thereby contribute to their wide geographical distribution (*Zeigler, 2014*). In sum, we have elucidated the structure and mechanism of the largest iron storage complex to date indicating that alternative systems exist across nature to address the critical problem of safe and dynamic iron storage.

# Materials and methods

## Key resources table

| Reagent type (species) or resource | Designation | Source or reference | Identifiers | Additional information |
| --- | --- | --- | --- | --- |
| Strain, strain background (*E. coli*) | MegaX DH10B T1R | Thermo Fischer Scientific | C640003 | Cloning strain |
| Strain, strain background (*E. coli*) | One Shot BL21 Star (DE3) | Thermo Fischer Scientific | C601003 | Expression strain |
| Strain, strain background (Geobacillus stearothermophilus) | ATCC 7953 | ATCC | ATCC 7953 | |

*Continued on next page*

*Continued*

| Reagent type (species) or resource | Designation | Source or reference | Identifiers | Additional information |
|---|---|---|---|---|
| Recombinant DNA reagent | pETDuet1 | EMD Millipore | 71146-3 | Expression vector |
| Sequence-based reagent | Codon-optimized IMEF cargo protein gene + encapsulin capsid protein gene containing overhangs for Gibson Assembly (oligonucleotide gBlock) | Integrated DNA Technologies (IDT) | based on accessions: WP_039238473.1; WP_03923847 | gttaagtataagaaggagatatacaATGAAGGAAGAACTGGATGCTTTCCATCAGATTTTCACTACGACCAAAGAGGCAATCGAACGTTTTATGGCGGATGCTGACCCCGGTCATTGAGAACGCGGAGGACGATCATGAGCGCCTGTATTATCATCATATCTACGAAGAGGAGGAGCAACGTCTGTCGCGCCTGGACGTTCTGATCCCACTGATCGAAAAGTTTCAAGATGAAACCGACGAAGGCCTCTTCTCCCCCTCCAACAACGCCTTTAACCGTCTGCTTCAGGAGCTGAATCTGGAAAAATTCGGTTTGCATAACTTTATCGAGCATGTTGACCTGGCCCTTTTTAGTTTCACCGACGAGGAACGCCAGACATTGCTTAAAGAACTGCGTAAAGATGCCTATGAAGGCTATCAGTATGTTAAAGAAAAACTGGCAGAAATTAACGCTCGTTTTGATCACGATTACGCAGACCCGCATGCGCACCATGATGAACACCGTGACCATCTTGCGGATATGCCCTCAGCGGGTTCATCGCACGAAGAAGTGCAGCCT<br>GTTGCACATAAAAAGAAAGGTTTCACGGTGGGTTCATTAA<br>TCCAGTAAATTTCGCTTAAATATTACCGCTAGCTCAAAAAG<br>GAGGAAAAGTGAATGAACAAAAGCCAACTTTATCCGGATT<br>CACCACTGACGGATCAGGACTTCAACCAATTAGAC-CAAACC<br>GTGATTGAGGCTGCTCGTCGTCAGCTGGTGGGTCGTCGCT<br>TCATTGAGTTATATGGCCCATTGGGGCGTGG<br>CATGCAGAGTGTCTTCAACGATATCTTCATGGAGTCTCATG<br>AAGCGAAAATGGACTTCCAGGGCAGCTTTG<br>ACACGGAGGTAGAGTCCTCCCGTCGTGTAAACTATACCATTCCG<br>ATGTTATATAAAGACTTCGTGCTTTACTGGCGCGATCTGGAAC<br>AGAGCAAGGCACTCGATATTCCGATCGACTTTTCAGTGGCAG<br>CGAACGCTGCCCGCGACGTTGCGTTCCTGGAAGATCA-GATGA<br>TTTTCCATGGAAGCAAAGAATTTGATATCCCGGGTCTGATGAA<br>CGTGAAAGGTCGCCTGACCCATCTGATTGGCAATTGGTATGAG<br>TCGGGTAACGCCTTTCAGGATATTGTGGAGGCCCGCAATAAAT<br>TACTCGAAATGAACCACAATGGCCCCATATGCTCTCGTGCTGT<br>CCCCGGAGCTGTACTCACTCTTA<br>CATCGTGTGCATAAAGACACGAATGTGCTGGAGATCGAACAC<br>GTGCGCGAGTTGATTACTGCTGGGGTTTTTCAGTCGCCTGTCC<br>TCAAAGGGAAAAGTGGTGTGATCGTAAACACCGGTCGCAACAAT<br>CTGGATTTGGCTATCTCGGAAGATTTTGAGACTGCATACCTGGG<br>CGAGGAAGGTATGAACCATCCCTTTCGCGTGTACGAGA-CAGTTG<br>TTCTGCGCATCAAACGCCCGGCGGCCATTTGTACTTTAATCGAT<br>CCGGAAGAATAAattaacctaggctgctgccaccgct |

*Continued on next page*

*Continued*

| Reagent type (species) or resource | Designation | Source or reference | Identifiers | Additional information |
|---|---|---|---|---|
| Sequence-based reagent | Codon-optimized IMEF cargo protein gene w/o TP + encapsulin capsid protein gene containing overhangs for Gibson Assembly (oligonucleotide gBlock) | Integrated DNA Technologies (IDT) | based on accessions: WP_039238473.1; WP_039238471 | gttaagtataagaaggagatatacaATGAAGGAAGAACTGGATGCTTT CCATCAGATTTTCACTACGACCAAAGAGGCAATCGAACGTTTTA TGGCGATGCTGACCCCGGTCATTGAGAACGCGGAGGACGATCAT GAGCGCCTGTATTATCATCATATCTACGAAGAGGAGGAGCAACGT CTGTCGCGCCTGGACGTTCTGATCCCACTGATCGAAAAGTTTCAA GATGAAACCGACGAAGGCCTCTTCTCCCCCTCCAACAACGCCTTT AACCGTCTGCTTCAGGAGCTGAATCTGGAAAAATT CGGTTTGCATAACTTTATCGAGCATGT TGACCTGGCCCTTTTTAGTTTCACCGACGAGGAACGCCAGACATTG CTTAAAGAACTGCGTAAAGATGCCTATGAAGGCTATCAGTATGTTA AAGAAAAACTGGCAGAAATTAACGCTCGTTTTGATCACGATTACGC AGACCCGCATGCGCACCATGATGAACACCGTGACCATCTTGCGGA TATGCCCTCAGCGGGTTCATCGCACGAAGAAGTGCAGCCTGTTGCA TAAATTTCGCTTAAATATTACCGCTAGCTCAAAAAGGAGGAAAAGTG AATGAACAAAAGCCAACTTTATCCGGATTCACCACTGACGGATCAG GACTTCAACCAATTAGACCAAACCGTGATTGAGGCTGCTCGTCGTCAGCTGGTGGGT CGTCGCTTCATTGAGTTATATGGCCCA TTGGGGCGTGGCATGCAGAGTGTCTTCAACGATATCTTCATGGAGT CTCATGAAGCGAAAATGGACTTCCAGGGC AGCTTTGACACGGAGGTAGAGTCCTCCCGTCGTGTAAACTATACCAT TCCGATGTTATATAAAGACTTCGTGCTTTACTGGCGCGATCTGGAAC AGAGCAAGGCACTCGATATTCCGATCGACTTTTCAGTGGCAGCGAAC GCTGCCCGCGACGTTGCGTTCCTGGAAGATCAGATGATTTTCCAT GGAAGCAAAGAATTTGATATCCCGGGTCT GATGAACGTGAAAGGTCGCCTGACCCATCTGATTGGCAATTGGTATGAG TCGGGTAACGCCTTTCAGGATATTGTGGAGGCCCGCAATAAATTACTCGAAATGAACCACAATGGCCCATAT GCTCTCGTGCTGTCCCCGGAGCTGTACTCACTCTTACATCGTGTGCATAAAGACACGAATGTGCTGGAGATCGAACACG TGCGCGAGTTGATTACTGCTGGGGGTTTTTCAGTCGCCTGTCCTCAA AGGGAAAAGTGGTGTGATCGTAAACACCGGTCGCAACCAATCTGGATT TGGCTATCTCGGAAGATTTTGAGACTGCATACCTGGGCGAGGAAGGTA TGAACCATCCCTTTCGCGTGTACGAGACAGTTGTTCTGCGCATCAAAC GCCCGGCGGCCATTTGTACTTTAATCGATCCGGAAGAATAA attaacctaggctgctgccaccgct |
| Commercial assay or kit | Gibson Assembly Master Mix | New England Biolabs | E2611L | |
| Commercial assay or kit | 14% Novex Tris-Glycine Gel | Thermo Fischer Scientific | XP00140BOX | |

*Continued on next page*

*Continued*

| Reagent type (species) or resource | Designation | Source or reference | Identifiers | Additional information |
|---|---|---|---|---|
| Commercial assay or kit | MIDAS screen | Molecular Dimensions | MD1–59 | Crystallization screen |
| Commercial assay or kit | Pierce Coomassie Plus (Bradford) Assay | Thermo Fischer Scientific | 23236 | Protein concentration determination4 |
| Chemical compound, drug | Isopropyl-β-D-thiogalactoside | Millipore Sigma | 10724815001 | |
| Chemical compound, drug | Lysozyme | Millipore Sigma | L6876 | |
| Chemical compound, drug | DNAse I | Millipore Sigma | 11284932001 | |
| Chemical compound, drug | Ni-NTA agarose resin | Qiagen | 30210 | |
| Chemical compound, drug | Polyethylene glycol 8000 | Millipore Sigma | 1546605 | |
| Chemical compound, drug | Uranyl formate | EMS | 22450 | |
| Chemical compound, drug | Formaldehyde 37% in water | Millipore Sigma | 252549 | |
| Chemical compound, drug | Glutaraldehyde 25% in water | Millipore Sigma | G5882 | |
| Chemical compound, drug | Picric acid | Millipore Sigma | 197378 | |
| Chemical compound, drug | Sodium cacodylate | Millipore Sigma | C0250 | |
| Chemical compound, drug | Uranyl acetate | EMS | 22400 | |
| Chemical compound, drug | Propylene oxide | Millipore Sigma | 82320 | |
| Chemical compound, drug | Epon | EMS | 14910 | |
| Chemical compound, drug | Glycolic acid | Millipore Sigma | 798053 | |
| Chemical compound, drug | Trisodium citrate | Millipore Sigma | S1804 | |
| Chemical compound, drug | Ammonium iron (II) sulfate | Millipore Sigma | F1543 | |
| Chemical compound, drug | ortho-phenylenediamine | Millipore Sigma | P9029 | |
| Chemical compound, drug | Hydrogen peroxide 30 % in water | Millipore Sigma | 216763 | |
| Software, algorithm | Genome Neighborhood Network Tool (GNT) | *Gerlt et al., 2015* | https://efi.igb.illinois.edu/efi-gnt/ | |
| Software, algorithm | blastp | NIH NCBI | https://blast.ncbi.nlm.nih.gov/Blast.cgi?PAGE=Proteins | |
| Software, algorithm | Clustal Omega | *McWilliam et al., 2013* | https://www.ebi.ac.uk/Tools/msa/clustalo/ | |

*Continued on next page*

*Continued*

| Reagent type (species) or resource | Designation | Source or reference | Identifiers | Additional information |
|---|---|---|---|---|
| Software, algorithm | Simply Phylogeny | *Madeira et al., 2019* | https://www.ebi.ac.uk/Tools/phylogeny/simple_phylogeny/ | |
| Software, algorithm | Geneious 9.14 | Biomatters Ltd | https://www.geneious.com/ | |
| Software, algorithm | UCSF Chimera 1.13 | *Pettersen et al., 2004* | https://www.cgl.ucsf.edu/chimera/ | |
| Software, algorithm | Open Source PyMOL | Schroedinger LLC | https://github.com/schrodinger/pymol-open-source | |
| Software, algorithm | I-TASSER | *Roy et al., 2010* | https://zhanglab.ccmb.med.umich.edu/I-TASSER/ | |
| Software, algorithm | IDT Codon Optimization Tool | Integrated DNA Technologies (IDT) | https://www.idtdna.com | |
| Software, algorithm | MotionCor2 | *Zheng et al., 2017* | https://omictools.com/motioncor2-tool | |
| Software, algorithm | CTFFIND4 | *Rohou and Grigorieff, 2015* | http://grigorieflab.janelia.org/ctffind4 | |
| Software, algorithm | SAMUEL | Liao Lab | https://liao.hms.harvard.edu/samuel | |
| Software, algorithm | Sam Viewer | Liao Lab | https://liao.hms.harvard.edu/samviewer | |
| Software, algorithm | Relion 3.0 | *Scheres, 2012* | https://www3.mrc-lmb.cam.ac.uk/relion/index.php?title=Main_Page | |
| Software, algorithm | SPIDER | *Frank et al., 1996* | https://spider.wadsworth.org/spider_doc/spider/docs/spider.html | |
| Software, algorithm | ResMap | *Swint-Kruse and Brown, 2005* | http://resmap.sourceforge.net/ | |
| Software, algorithm | Coot 0.8.9.1 | *Emsley et al., 2010* | https://www2.mrc-lmb.cam.ac.uk/personal/pemsley/coot/ | |
| Software, algorithm | Phenix 1.14 | *Adams et al., 2010* | http://www.phenix-online.org/ | |
| Software, algorithm | XDS | *Kabsch, 2010* | http://xds.mpimf-heidelberg.mpg.de/ | |
| Software, algorithm | ACRIMBOLDO_LITE | *Sammito et al., 2015* | http://chango.ibmb.csic.es/arcimboldo_lite | |
| Software, algorithm | Phaser | *McCoy et al., 2007* | https://www.phaser.cimr.cam.ac.uk/index.php/Phaser_Crystallographic_Software | |

*Continued on next page*

*Continued*

| Reagent type (species) or resource | Designation | Source or reference | Identifiers | Additional information |
|---|---|---|---|---|
| Software, algorithm | SHELX | *Thorn and Sheldrick, 2013* | http://shelx.uni-goettingen.de/ | |
| Software, algorithm | CCP4 | *Winn et al., 2011* | http://www.ccp4.ac.uk/ | |
| Software, algorithm | REFMAC5 | *Murshudov et al., 1997* | http://www.ccp4.ac.uk/html/refmac5.html | |
| Software, algorithm | Fiji-ImageJ 1.52h | *Schindelin et al., 2012* | https://fiji.sc/ | |
| Software, algorithm | UCSFImage4 | omicX | https://omictools.com/ucsfimage-tool | |
| Other | 200 Mesh Gold Grids | EMS | FCF-200-Au | |
| Other | 400 Mesh Cu Holy Carbon Grids | EMS | Q410CR1.3 | |

## Computational analysis of genomes, encapsulin gene clusters, sequences and protein structures

Initial identification of IMEF-systems was achieved by utilizing the Enzyme Similarity Tool (ESI) in combination with the Genome Neighborhood Network Tool (GNT) of the Enzyme Function Initiative (EFI) (*Gerlt et al., 2015*). The previously identified IMEF cargo protein from *Q. thermotolerans* (WP_039238471) was used as a query to initiate an ESI Sequence BLAST search of the UniProt database. UniProt BLAST Query E-value was chosen to be 5. After the initial dataset was created, we used an alignment score (based on the alignment score vs percent identity plot) that would correspond to a percent identity of 20 for initial outputting and interpretation of protein sequences and sequence similarity networks (SSNs). The resulting xgmml network file was then submitted to GNT. The resulting Genome Neighborhood Diagrams of all identified IMEF operons where analyzed using the GNT diagram explorer and operon diagrams were downloaded as svg files.

Genomes of IMEF-system-encoding organisms were searched for Ftn, Bfr and Dps proteins using NCBI's blastp suite. As queries, Firmicute homologs of ferritin, bacterioferritin and Dps were used (Ftn: OTY20392, Bfr: EEK74551, Dps: WP_039234032).

Phylogenetic analysis was based on Clustal Omega (ClustalO) alignments carried out using the default settings of the Multiple Sequence Alignment online tool of the European Molecular Biology Laboratory's European Bioinformatics Institute (EMBL-EBI). A nearest-neighbor phylogenetic trees based on the ClustalO alignment were generated using the Simple Phylogeny Tool at EMBL-EBI. Alignments and trees were then annotated and analyzed using Geneious 9.1.4.

Cryo-EM data and structural models were analyzed using UCSF Chimera 1.13.1rc and Open Source PyMOL 1.8.x. Structural alignments of capsid protein monomers were carried out in PyMOL using the align command. The IMEF model used for molecular replacement was generated using the I-TASSER webserver (*Roy et al., 2010*; *Yang and Zhang, 2015*).

## Molecular biology and cloning

All constructs used in this study were ordered as gBlock Gene Fragments from Integrated DNA Technologies (IDT). Codon usage was optimized for *E. coli* expression using the IDT Codon Optimization Tool with the amino acid sequences of the respective proteins of interest as input. For the IMEF operon containing multiple genes, intergenic regions were not changed. The IMEF cargo protein construct was ordered with a C-terminal His$_6$ tag. For the operon construct containing the TP-less IMEF cargo, the 13 C-terminal residues (HKKKGFTVGSLIQ) were omitted from the IMEF cargo protein, thus removing the TP.

Gibson Assembly Master Mix was obtained from New England BioLabs (NEB). DNA sequencing was carried out by GENEWIZ. MegaX DH10B T1R electrocompetent *E. coli* cells (ThermoFisher) were used for all cloning procedures while One Shot BL21 Star (DE3) chemically competent *E. coli*

cells (Invitrogen) were used for protein production and all other experiments. pETDuet1 was used as the expression vector for all constructs. For the construction of expression vectors, Gibson Assembly was employed. gBlock Gene Fragments containing 20 bp overlaps for direct assembly were combined with NdeI and PacI digested pETDuet1 resulting in assembled expression vectors (fragments were inserted in MCS2). Electrocompetent *E. coli* DH10B cells were transformed and the resulting plasmids confirmed via sequencing.

## Expression and purification of proteins and protein compartments

All non-high iron expression experiments were carried out in lysogeny broth (LB) supplemented with ampicillin (100 µg/mL). Size exclusion chromatography/gel filtration for capsid purification was performed with an ÄKTA Explorer 10 (GE Healthcare Life Sciences) equipped with a HiPrep 16/60 Sephacryl S-500 HR column (GE Healthcare Life Sciences). For analytical size exclusion, a Superdex 200 10/300 GL column (GE Healthcare Life Sciences) was used. Protein samples were concentrated using Amicon Ultra Filters (Millipore). For SDS-PAGE analysis, 14% Novex Tris-Glycine Gels (ThermoFisher Scientific) were used. DNA concentrations were measured using a Nanodrop ND-1000 instrument (PEQLab).

Sequence-confirmed plasmids were used to transform *E. coli* BL21 (DE3) Star cells (0.5 ng total plasmid DNA). Resulting colonies were used to inoculate pre-expression cultures.

For large scale protein expressions, 500 mL of LB in 2 L baffled flasks were inoculated (1:50) using an over-night culture, grown at 37°C and 200 rpm to an $OD_{600}$ of 0.5. The temperature was then shifted to 30°C and the cultures induced with IPTG (final concentration: 0.05 mM). Cultures were grown at 30°C for 18 hr, harvested through centrifugation (4000 rpm, 15 min, 4°C) and pellets either immediately used or frozen in liquid nitrogen and stored at −20°C for later use.

For encapsulin and His-tagged protein purifications, pellets were thawed, resuspended in 5 mL Tris buffer (50 mM Tris, 150 mM NaCl, pH 8), then lysozyme (1 mg/mL) and DNaseI (1 µg/mL) were added and the cells incubated on ice for 20 min. Cell suspensions were subjected to sonication using a 550 Sonic Dismembrator (FisherScientific). Power level 3.25 was used with a pulse time of 8 s and an interval of 10 s. Total pulse time was 4 min. Cell debris was subsequently removed through centrifugation (8000 rpm, 15 min, 4°C). The cleared supernatant was then used either for protein affinity or encapsulin compartment purification.

His-tagged IMEF cargo was purified using Ni-NTA agarose resin (Qiagen) via the batch Ni-NTA affinity procedure following the supplier's instructions. Buffer A (50 mM Tris, 150 mM NaCl, 20 mM imidazole, pH 8) was used to wash the resin after protein binding and buffer B (50 mM Tris, 150 mM NaCl, 250 mM imidazole, pH 8) was used to elute bound protein. Samples were concentrated and dialyzed using Amicon filters (10 kDa molecular weight cutoff) and Tris (pH 7.4) buffer and evaluated using SDS-PAGE. Further analyses were carried out directly or the next day with protein being stored on ice.

For encapsulin purification, 0.1 g NaCl and 0.5 g of PEG-8000 were added (10% w/v final concentration) to 5 mL cleared lysate, followed by incubation on ice for 20 min. The precipitate was collected through centrifugation (8000 rpm, 15 min, 4°C), suspended in 3 mL Tris (pH 8) buffer and filtered using a 0.2 µm syringe filter. The samples were then subjected to size exclusion chromatography using Tris (pH 8) buffer and a flow rate of 1 mL/min.

Fractions were evaluated using SDS-PAGE analysis and encapsulin-containing fractions were combined, concentrated and dialyzed using Amicon filters (100 kDa molecular weight cutoff) and Tris buffer without NaCl (20 mM Tris, pH 8).

The low salt sample was then loaded on a HiPrep DEAE FF 16/10 Ion Exchange column (GE Healthcare Life Sciences). The gradient used for ion-exchange chromatography was as follows: 100% A for 0–100 mL, 100% A to 50% A + 50% B for 100–200 mL, 100% B for 200–300 mL, 100% A for 300–400 mL (A: 20 mM Tris, pH 8, B: 20 mM Tris, 1 M NaCl, pH 8, flow rate: 3 mL/min). Again, SDS-PAGE was used to identify product fractions followed by Amicon filter concentration and buffer exchange to Tris buffer (50 mM Tris, 150 mM NaCl, pH 8).

Final samples were either directly subjected to additional experiments or stored on ice overnight.

## Negative stain transmission electron microscopy (TEM) of purified encapsulins

200 Mesh Gold Grids (FCF-200-Au, EMS) were used for all negative stain TEM experiments. TEM experiments of negatively stained protein samples were carried out at the HMS Electron Microscopy Facility using a Tecnai G2 Spirit BioTWIN instrument.

For negative-staining TEM, encapsulin samples were diluted to 1–10 µM using Tris buffer (50 mM Tris, 150 mM NaCl, pH 8) and subsequently adsorbed onto formvar/carbon coated gold grids. Prior to applying 5 µL of diluted sample, grids were glow-discharged using a 100x glow discharge unit (EMS) to increase their hydrophilicity (10 s, 25 mA). After 1 min adsorption time, excess liquid was blotted off using Whatman #1 filter paper, washed one time with distilled $H_2O$ and floated on a 10 µL drop of staining solution (0.75% uranyl formate in $H_2O$) for 35 s. After removal of excess staining solution, samples were used for TEM analysis at 80 kV.

## Thin section TEM analysis of fixed bacterial cells

For TEM analysis of fixed cells, 0.5 mL of early stationary phase bacterial culture was fixed by adding fixative (1:1 v/v, 1.25% formaldehyde, 2.5% glutaraldehyde, 0.03% picric acid in 0.1 M sodium cacodylate buffer, pH 7.4). The sample was then incubated at 25°C for 1 hr and centrifuged for 3 min at 3000 rpm. The sample was then further incubated for 6–18 hr at 4°C. Cells were subsequently washed three times in cacodylate buffer, 4 times with maleate buffer pH 5.15 followed by staining with 1% uranyl acetate for 30 min. The sample was dehydrated (15 min 70% ethanol, 15 min 90% ethanol, 2 × 15 min 100% ethanol) and exposed to propyleneoxide for 1 hr. For infiltration, a mixture of Epon resin and proylenoxide (1:1) was incubated for 2 hr at 25°C before moving it to an embedding mold filled with freshly mixed Epon. The sample was allowed to sink and subsequently moved to a polymerization oven (24 hr, 60°C). Ultrathin sections (60–90 nm) were then cut at −120°C using a cryo-diamond knife (Reichert cryo-ultramicrotome) and transferred to formvar/carbon coated grids.

## Cryo-electron microscopy (cryo-EM) data collection and processing

To prepare grids for cryo-EM imaging, 2.5 µL of purified cargo-loaded IMEF encapsulin at a concentration of 1.5 mg/mL was applied to glow-discharged Quantifoil holey carbon grids (1.2/1.3, 400 mesh), and blotted for 3 s with ~90% humidity before plunge-freezing in liquid ethane using a Cryoplunge 3 System (CP3, Gatan). Cryo-EM images were collected at Harvard Medical School on a Tecnai F20 electron microscope (FEI) operating at 200 kV and equipped with a K2 Summit direct electron detector (Gatan). Movies were collected at a nominal magnification of 29,000 with a calibrated pixel size of 0.64 Å. All movies were collected in super-resolution counting mode using UCSFImage4, with a total exposure time of 7.2 s and a frame time of 200 milliseconds. The details of EM data collection parameters are listed in *Supplementary file 2*.

Dose-fractionated super-resolution movies collected on the K2 detector were binned over 2 × 2 pixels, and subjected to motion correction using the program MotionCor2 (*Zheng et al., 2017*). Dose-weighted sums from all frames were used for all subsequent image-processing steps except for defocus determination. The CTFFIND4 program (*Rohou and Grigorieff, 2015*) was used to determine the defocus values of the summed images from all movie frames without dose weighting. Semi-automated particle picking from 6x binned images was performed with SAMUEL and SamViewer (*Ru et al., 2015*). Selected particles were extracted from unbinned images with an initial box size of 512 pixels, and subsequently binned to a box size of 128 pixels with a pixel size of 5.12 Å for two rounds of 2D classification using RELION 3.0 (*Scheres, 2012*). An initial 3D model was generated via SPIDER (*Frank et al., 1996*) 3D projection matching refinement (samrefine.py) using 2D class averages, starting from a sphere density similar in size and shape of the IMEF encapsulin. The selected particles after 2D classification were binned to a box size of 480 pixels (corresponding to a pixel size of 1.365 Å) and used for 3D refinement in RELION 3.0 with icosahedral symmetry ('I') imposed. A final round of 3D refinement was performed in RELION 3.0 after fitting individual particle defocus parameters and beam-tilt with 'relion_ctf_refine'. Post-processing was performed with 'relion_postprocess' to apply a negative b-factor and correct the amplitude information in the final map. The overall resolutions were estimated based on the gold-standard criterion of Fourier shell

correlation (FSC) = 0.143. Local resolution variations were estimated from two half data maps using ResMap (*Swint-Kruse and Brown, 2005*).

## Cryo-EM model building and refinement

An initial model of an IMEF encapsulin monomer was generated by homology modeling with the I-TASSER webserver (*Zhang, 2008*) using the x-ray crystal structure of the T = 3 *Pyrococcus furiosus* encapsulin (PDB ID: 2E0Z) as a template. The monomer model was then fit into the 3D map in UCSF Chimera (*Pettersen et al., 2004*), and subsequently adjusted manually in COOT (*Emsley et al., 2010*) prior to refinement in PHENIX (*Adams et al., 2010*) with phenix.real_space_refine. The refined monomer coordinates were copied and manually positioned to occupy the four monomer positions of the asymmetric unit (ASU), followed by manual adjustment of each monomer in COOT. Several rounds of real-space refinement and manual adjustment of the coordinates for four monomers in the ASU were performed in phenix.real_space_refine and COOT. During refinement of coordinates in the ASU no non-crystallographic symmetry restraints were utilized in order to avoid distortion of the E-loop in each monomer. The refined coordinates for the ASU were subsequently expanded using the symmetry matrices utilized by RELION 3.0 during 3D reconstruction to generate a model of the entire encapsulin cage containing 60 ASUs and 240 total IMEF encapsulin capsid protein polypeptide chains. Coordinates for the entire IMEF encapsulin cage were refined in phenix.real_space_refine with proper NCS restraints between corresponding chains in individual ASUs in order to resolve any inter-protomer clashes.

## Symmetry expansion and focused classification

In an attempt to better resolve cargo density within the encapsulin shell we used an approach combining symmetry expansion and focused classification with residual signal subtraction. Prior to symmetry expansion and focused classification, particles were binned to a box size of 192 with a corresponding pixel size of 3.41 Å. Following refinement of binned particles with icosahedral symmetry, a 60 Å low-pass filtered mask of a hexameric encapsulin shell unit with associated cargo density was generated (*Figure 1—figure supplement 3a*). Symmetry expansion was performed with relion_particle_symmetry_expand specifying 'I' symmetry to generate a new particle stack with 60x increased particle number. Residual signal subtraction was performed as described previously (*Bai et al., 2015*) to subtract encapsulin shell and cargo densities outside of the 60 Å low-pass filtered mask from the symmetry expanded particle dataset (*Figure 1—figure supplement 3b*). Focused classification without alignment and without applied symmetry was then performed in Relion3.0 to resolve cargo density bound in different configurations to the encapsulin shell and potential connections between the cargo and targeting peptide (*Figure 1—figure supplement 3c*).

## Differential scanning fluorimetry (DSF) to test thermal stability of proteins

DSF measurements were performed using a NanoTemper Tycho NT.6 instrument according to the manufacturer's instructions. Samples in Tris buffer (50 mM Tris, 150 mM NaCl, pH 8) at a concentration of 0.5 mg/mL were measured in triplicate and subjected to a temperature gradient from 35°C to 95°C at 0.5°C per second. Data were analyzed using NT Melting Control software. Melting temperatures (Tm) were determined by automatic fitting of experimental data using a polynomial function, where the maximum slope (Tm) is indicated by the peak of its first derivative.

## Crystallization and x-ray structure determination of the IMEF cargo protein

Initial crystallization conditions were determined using the Midas screen (*Grimm et al., 2010*). Large single crystals were grown in sitting drop plates by the vapor diffusion method. Reservoir solutions contained 10% v/v Pentaerythritol ethoxylate (3/4 EO/OH) and 10% butanol. Crystals were cryo-protected in reservoir solution supplemented with 15% ethylene glycol and 20 mM glycolic acid pH 7.5. Diffraction data were collected at the European Synchrotron Radiation Facility (ESRF) Grenoble outstation at the ID-30b beamline at 100 K with a Pilatus3 6M pixel detector (DECTRIS, Switzerland). Data were indexed, processed, and scaled with the XDS package (*Kabsch, 2010*). The structure was solved by molecular replacement using an I-TASSER homology model and the program

ACRIMBOLDO_LITE (*Sammito et al., 2015*) incorporating PHASER (*McCoy et al., 2007*) and SHELX (*Thorn and Sheldrick, 2013*) from the CCP4 suite (*Winn et al., 2011*). Model building and refinement was carried using COOT (*Emsley and Cowtan, 2004*) and REFMAC5 (*Murshudov et al., 1997*),respectively.

### Determination of electron-dense core diameters

To determine the size distribution of electron-dense cores resulting from IMEF mineralization under high iron conditions, TEM micrographs were analyzed using the open source image processing package Fiji based on ImageJ 1.52 hr (*Schindelin et al., 2012*). Micrographs were converted to 8-bit binary images, thresholded and processed using the particle analyzer plugin. The diameters reported are based on Fiji Feret diameter output values.

### In vivo mineralization of electron-dense particles

Overnight cultures were used to inoculate 500 mL LB medium (1:50) supplemented with ampicillin and grown at 37°C to an $OD_{600}$ of 0.5. Expression was induced with 0.05 mM IPTG. Cultures were incubated at 30°C for 2 hr. LB medium was removed and replaced with fresh modified LB (LB +50 mM Hepes, 4 mM Trisodium citrate, pH 7) supplemented with freshly prepared ammonium iron(II) sulfate ($Fe(NH_4)_2(SO_4)_2$, final concentration: 4 mM; stock solution: 400 mM in 0.1 M HCl). The cultures were then incubated at 30°C for 18 hr and used for either the purification of iron-loaded encapsulin compartments or thin section TEM analysis.

### Iron-rich core characterization via energy-dispersive x-ray spectroscopy (EDS) and electron energy loss spectroscopy (EELS)

TEM and high angle angular dark field (HAADF) STEM imaging and analysis were performed on a JEOL ARM 200F operated at 80 kV. EDS spectra were collected using an EDAX Octane W 100mm2 detector, and spectra analyzed post-collection both via TEAM software and offline using the k-ratio method (thin film approximation). EELS mapping data of the Fe L edge were acquired using a Gatan Enfinium spectrometer with dispersion 0.25 eV/ch using DualEELS mode with simultaneous zero loss spectrum collection. EELS data were processed using the Gatan EELS analysis plug-in. The processing steps involved a Gaussian fitting of the zero loss peak, integrating under the FeL edge up to 780 eV after applying a power law or first order log-polynomial (whichever fit the background better, as this depended on local carbon contamination levels) and correcting for the Fe cross section of 2664.9 barns, from which the average number of Fe per $nm^2$ was calculated per pixel of data. These pixels were summed over the area of each particle to estimate the total number of Fe atoms. Errors in this measurement were calculated from a statistical analysis of the data fitting combined with the expected error from Fe cross sectional extrapolation. Particle diameters were estimated using a histogram method to determine the edge onset of each particle, with the mean of multiple measurements from each particle used (and error determined by the standard deviation of these measurements).

### Cultivation of *Geobacillus stearothermophilus* ATCC 7953

For normal growth of *G. stearothermophilus*, Meat Media (3 g meat extract, 5 g peptone, 1 L $H_2O$) was utilized. *G. stearothermophilus* was maintained on Meat Media agar plates (15 g agar/L). All growth was carried out a 55°C. For high iron growth experiments Meat Media was supplemented with 50 mM Hepes, 4 mM Trisodium citrate and 4 mM $Fe(NH_4)_2(SO_4)_2$ and the pH adjusted to seven using HCl. Growth curves were recorded in high iron Meat Media in 96-well plates (volume: 500 μL) using a Synergy H1 plate reader (BioTek) and inoculated (1:50) from a pre-culture grown for 24 hr in standard Meat Media.

### Peroxidase assays

Peroxidase activity of free IMEF cargo and cargo-loaded IMEF encapsulin was assayed by measuring the oxidation of *ortho*-phenylenediamine (OP) by hydrogen peroxide (*Pesek et al., 2011*). OP dilutions from 10 to 80 mM were prepared from a stock solution (92.5 mM in 50 mM Tris, pH 8) using Tris buffer (pH 8). 96-well plates were used to carry out the assays in triplicate. Each well contained 100 μL of OP dilution and 0.5 μM of IMEF cargo protein (protein concentrations were determined

via Bradford assay (Pierce Coomassie, ThermoFisher) following the manufacturer's instructions). To start the assays, 2 µL of 30% hydrogen peroxide solution was added. After 15 min of incubation in the dark, assays were stopped by the addition of 100 µL of 0.5 M $H_2SO_4$. Then, absorbance at 490 nm was determined using a Synergy H1 plate reader.

## Ferroxidase assays

Protein solutions in Tris buffer (50 mM Tris, 150 mM NaCl, pH 8) and $Fe(NH_4)_2(SO_4)_2$ stock solutions in 0.1 M HCl were made anaerobic by incubation in a Vinyl Anaerobic Chamber (Coy Lab Products) for 24 hr. All solutions were exposed to the anaerobic atmosphere inside the chamber and protein solutions were kept on ice. IMEF cargo protein was used at a final concentration of 50 µM while cargo-loaded encapsulin concentrations were used that would correspond to 5 µM IMEF cargo (higher concentrations led to rapid protein precipitation upon iron addition). Final iron(II) concentrations ranged from 10 to 100 µM. Ferroxidase activity was initiated by combining appropriate dilutions of protein and iron solution to a final volume of 250 µL in a quartz cuvette in the air, directly after removing solutions from the anaerobic chamber. Ferroxidase activity was immediately measured by monitoring $Fe^{3+}$ formation at a wavelength of 315 nm in a Nanodrop 2000c for 25 min.

## Data availability

A cryo-EM density map of the cargo-loaded IMEF encapsulin has been deposited in the Electron Microscopy Data Bank under the accession number 9383. The corresponding atomic coordinates for the atomic model have been deposited in the Protein Data Bank (accession number: 6NJ8). Atomic coordinates for the IMEF cargo protein have been deposited in the Protein Data Bank under accession number 6N63. Correspondence and requests for materials should be addressed to the corresponding authors.

# Acknowledgements

We thank M Ericsson, P Coughlin and L Trakimas for technical support (HMS Electron Microscopy Facility). This work was supported by a Leopoldina Research Fellowship (LPDS 2014–05) from the German National Academy of Sciences Leopoldina (TWG), the Wyss Institute for Biologically Inspired Engineering (TWG and PAS) and the Gordon and Betty Moore Foundation (TWG and PAS, Grant Number 5506).

# Additional information

## Funding

| Funder | Grant reference number | Author |
| --- | --- | --- |
| Deutsche Akademie der Naturforscher Leopoldina - Nationale Akademie der Wissenschaften | LPDS 2014-05 | Tobias W Giessen |
| Gordon and Betty Moore Foundation | 5506 | Tobias W Giessen Pamela A Silver |
| Wyss Institute for Biologically Inspired Engineering | | Tobias W Giessen Pamela A Silver |

The funders had no role in study design, data collection and interpretation, or the decision to submit the work for publication.

## Author contributions

Tobias W Giessen, Conceptualization, Formal analysis, Supervision, Investigation, Methodology, Writing—original draft, Project administration, Writing—review and editing; Benjamin J Orlando, Data curation, Formal analysis, Methodology, Writing—review and editing; Andrew A Verdegaal, Melissa G Chambers, Investigation, Methodology; Jules Gardener, David C Bell, Formal analysis, Methodology; Gabriel Birrane, Data curation, Formal analysis, Methodology; Maofu Liao, Formal

analysis, Methodology, Writing—review and editing; Pamela A Silver, Conceptualization, Resources, Supervision, Writing—review and editing

### Author ORCIDs

Tobias W Giessen  https://orcid.org/0000-0001-6328-2031
Andrew A Verdegaal  https://orcid.org/0000-0002-4517-6961
Melissa G Chambers  https://orcid.org/0000-0001-5111-7194
Gabriel Birrane  http://orcid.org/0000-0002-1759-5499

### Decision letter and Author response

Decision letter https://doi.org/10.7554/eLife.46070.031
Author response https://doi.org/10.7554/eLife.46070.032

## Additional files

### Supplementary files

• Supplementary file 1. Table of ferritin-like proteins (Flps) identified in IMEF operon-containing Firmicutes. All identified ferritins (Ftn), bacterioferritins (Btf) and DNA-binding proteins from starved cells (Dps) found in IMEF operon strains. IMEF and encapsulin capsid protein IDs are shown as well. 97% of IMEF operon-containing strains do not encode Ftn, 93% do not encode Bfr and 92% do not encode either. However, 93% of IMEF operon-encoding strains encode Dps systems. This likely indicates that the encapsulin based IMEF system represents the major iron storage system in 92% of the listed strains. It also indicates that IMEF systems do likely not function as unusual Dps system given that the vast majority of strains encode standard Dps system. Blast searches were carried out using the NCBI Blastp server with the following sequences as queries: Ftn: OTY20392, Bfr: EEK74551, Dps: WP_039234032, IMEF: WP_039238473, Encapsulin: WP_039238471.
DOI: https://doi.org/10.7554/eLife.46070.018

• Supplementary file 2. Table of cryo-EM data collection statistics for IMEF-loaded encapsulin.
DOI: https://doi.org/10.7554/eLife.46070.019

• Supplementary file 3. List of ferritin-like protein and IMEF protein IDs used to construct the phylogenetic tree shown in *Figure 3A*. IMEF: Iron-mineralizing encapsulin-associated Firmicute cargo, EncFlp: ferritin-like proteins (Flps) found within encapsulin operons containing targeting peptides, noEncFlp: Flps found outside encapsulin operons not containing a targeting peptide, Bfr: bacterioferritin, Rr: rubrerythrin, Mam-Ftn: mammalian ferritin, Bac-Ftn: bacterial ferritin, Dps: DNA-binding proteins from starved cells.
DOI: https://doi.org/10.7554/eLife.46070.020

• Supplementary file 4. X-ray structure determination and refinement statistics for the IMEF cargo protein.
DOI: https://doi.org/10.7554/eLife.46070.021

• Supplementary file 5. EELS data of electron dense cores of purified IMEF encapsulins produced in *E. coli* under high iron conditions. #Fe from EELS analysis values were extracted using Gatan software. Errors combine estimated statistical error in this measurement with known error for cross section. For density calculations, particles were approximate as spheres.
DOI: https://doi.org/10.7554/eLife.46070.022

• Transparent reporting form
DOI: https://doi.org/10.7554/eLife.46070.023

### Data availability

A cryo-EM density map of the cargo-loaded IMEF encapsulin has been deposited in the Electron Microscopy Data Bank under the accession number 9383. The corresponding atomic coordinates for the atomic model have been deposited in the Protein Data Bank (accession number: 6NJ8). Atomic coordinates for the IMEF cargo protein have been deposited in the Protein Data Bank under accession number 6N63.

The following datasets were generated:

| Author(s) | Year | Dataset title | Dataset URL | Database and Identifier |
|---|---|---|---|---|
| Maofu L, Giessen TW, Silver PA, Orlando BJ | 2019 | Encapsulin iron storage compartment from Quasibacillus thermotolerans | http://www.ebi.ac.uk/pdbe/entry/emdb/EMD-9383 | EMDataBank, EMD-9383 |
| Maofu L, Giessen TW, Silver PA, Orlando BJ | 2019 | Encapsulin iron storage compartment from Quasibacillus thermotolerans | http://www.rcsb.org/structure/6NJ8 | Protein Data Bank, 6NJ8 |
| Giessen TW, Birrane G | 2019 | Crystal structure of an Iron binding protein | http://www.rcsb.org/structure/6N63 | Protein Data Bank, 6N63 |

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
