## [Decision Letter]

Thank you for submitting your article "Large protein organelles form a new iron sequestration system with high storage capacity" for consideration by *eLife*. Your article has been reviewed by Gisela Storz as the Senior Editor, a Reviewing Editor, and three reviewers. The following individual involved in review of your submission has agreed to reveal his identity: Robert Crichton (Reviewer #1).

The reviewers have discussed the reviews with one another and the Senior Editor has drafted this decision to help you prepare a revised submission.

Summary:

This manuscript presents a structural and mechanistic description of a recently discovered bacterial iron storage system based on a T4 icosahedral encapsulin shell carrying an iron-mineralising protein cargo of the iron mineralizing encapsulin-associated firmicute (IMEF) family. The study is sound and well communicated, and the findings are significant. The authors present a single-particle cryo-EM study of the encapsulin shell, and an X-ray crystal structure of the IMEF protein, which forms a ferroxidase center at its dimeric interface. A target peptide of the IMEF protein is shown to be essential to incorporation of the IMEF cargo into the encapsulin shell. The chemical composition of heterologously-expressed Enc-IMEF is analysed by EDS and EELS, and the ferroxidase activity is analysed to indicate that iron mineralisation by the full complex is limited by the rate of iron entry to the encapsulin shell.

Essential revisions:

Most of the comments pertain to improving the presentation and discussion of the work.

1) There seems to be a systemic problem with not introducing or explaining key ideas and relationships early enough. Some of the confusing points become partly clarified in the end after multiple readings. But for someone not well-acquainted with these systems, and frankly even for those closer to the subject, one is left without a clear sense of how different the findings and the specific structure should be considered compared to other studies on triangulated encapsulins and even HK97 phage capsids. The function, composition, equivalence or distinction compared to various encapsulins, is hard to absorb; could other encapsulins be doing this and it just hasn't been shown.

2) The authors state that 'A newly discovered class of protein organelles called encapsulin nanocompartments are implicated in microbial iron and redox metabolism and have so far only been shown to be involved in oxidative stress response (Giessen and Silver, 2017; He et al., 2016; McHugh et al., 2014; Sutter et al., 2008).', but cite a paper (McHugh, 2014) in which iron storage by an encapsulin is well documented. The authors' own work (Giessen and Silver, 2017) has previously shown that IMEF-Enc mineralises iron in vivo. I would therefore consider it established that encapsulins can function in iron storage.

3) Other issues related to clarity:

- Is IMEF a system or is IMEF a cargo protein?

-Retention of the 'cargo protein' name instead of a protein name based on homology and presumptive function allows questions to linger unnecessarily.

- The main statement about what protein construct/assembly is produced for study is (subsection “Overall structure of the cargo-loaded IMEF encapsulin”) "we produced homogeneous IMEF cargo-loaded encapsulins". What does that mean? What proteins were expressed?

- Subsection “Overall structure of the cargo-loaded IMEF encapsulin”: "as evidenced by comparison of the IMEF T =4 monomer with T = 1, T = 3 and T = 7 capsid proteins." What capsid proteins? Is this referring to all encapsulin and HK97 proteins or something else?

- There are places where "the" should probably be "a" instead, where a new idea hasn't been introduced previously. [subsection “Overall structure of the cargo-loaded IMEF encapsulin” on the flexibility of a linker in the cargo protein].

- Subsection “Structure and analysis of the IMEF cargo protein” says that a phylogeny analysis shows IMEF is a member of the Flp superfamily, but could not be detected as the sequence level. What is meant here? That the IMEF protein has sequence similarity to other proteins whose structures were known and could be assigned to the Flp superfamily despite not being able to detect sequence similarity to other Flp members?

A few technical issues also need to be addressed:

4) The following points can be addressed changes to the text:

- The issue of symmetry averaging and its presumptive effects on certain parts of the structure like the cargo are not handled cleanly (See subsection “Overall structure of the cargo-loaded IMEF encapsulin”; subsection “TP-mediated cargo-shell co-assembly”). The authors infer flexibility in some cases where lack of icosahedral symmetry in the presence of averaging would likely have the same effect. How would the cargo protein survive averaging if it sits as a single dimer bound to a pentamer at an icosahedral vertex?

- In subsection “Iron mineralization and storage by the IMEF system”, the logic about the shell permeability and kinetic curve shapes is unclear.

- In subsection “Iron mineralization and storage by the IMEF system”, the idea of being "channeled to pores" is contrasted with diffusion in the next phrase. But presumptive pore transport here is presumably diffusive. The physical ideas need to be spelled out more carefully.

- Grounds are lacking for the assertion in subsection “Overall structure of the cargo-loaded IMEF encapsulin” about the observed conformational diversity being important for pore function.

- In subsection “Non-covalent chainmail and thermal stability of the IMEF system”, the absence of a patent pore is not evidence for a gated pore.

- More caution is required on the claim of ions and density in the central regions of the capsid oligomers. For one, averaging often accentuates noise on symmetry axes. But further the identities/charge of any molecules there are entirely unknown; the densities could be water for example.

5) The authors discuss the probability that features of the cryo-EM map, including the IMEF densities, are artifacts of averaging, which is almost certainly the case. This could be mitigated by symmetry expansion (relion_particle_symmetry_expand) and focussed classification/refinement for a clearer picture of the IMEF protein within the encapsulin shell. This is not essential, but would strengthen the paper considerably.

---

## [Author Response]

Essential revisions:Most of the comments pertain to improving the presentation and discussion of the work.1) There seems to be a systemic problem with not introducing or explaining key ideas and relationships early enough. Some of the confusing points become partly clarified in the end after multiple readings. But for someone not well-acquainted with these systems, and frankly even for those closer to the subject, one is left without a clear sense of how different the findings and the specific structure should be considered compared to other studies on triangulated encapsulins and even HK97 phage capsids. The function, composition, equivalence or distinction compared to various encapsulins, is hard to absorb; could other encapsulins be doing this and it just hasn't been shown.

In our initial submission we tried to incorporate a reasonable amount of discussion and comparison with other systems. However, we agree that for scientists not familiar with the field, additional explanation and discussion will be very helpful. Therefore, we have included a substantial amount of additional information throughout the manuscript, particularly comparing and contrasting our findings with other encapsulin systems. We have included the following sentences:

Introduction:

“So far, operons involved in hydrogen peroxide and nitric oxide detoxification as well as iron mineralization have been reported (Nichols, Cassidy-Amstutz, Chaijarasphong and Savage, 2017). The main cargo protein-types described to date are DyP-type peroxidases, hemerythrins and different classes of ferritin-like proteins (Contreras et al., 2014; Giessen and Silver, 2017; McHugh et al., 2014; Rahmanpour and Bugg, 2013).”

Subsection “Overall structure of the cargo-loaded IMEF encapsulin “:

“The IMEF compartment is substantially larger than previously reported encapsulins and possesses a triangulation number of T = 4 instead of T = 1 (60 subunits, 24 nm) or T = 3 (180 subunits, 32 nm) and represents the largest encapsulin compartment reported to date (Figure 1—figure supplement 2E).”

Subsection “Overall structure of the cargo-loaded IMEF encapsulin”:

“In contrast, T = 1 encapsulins consist of only 12 pentameric capsomers while the T = 3 encapsulin shell is made up of 12 pentameric and 20 hexameric capsomers. The T = 4 IMEF-system consequently possesses an internal volume 530% and 220% larger than that of T = 1 and T = 3 encapsulins, respectively.”

Subsection “Pores in the IMEF encapsulin shell”:

“Similarly, pores at the symmetry axes were also reported for T = 1 and T = 3 encapsulin systems.”

Subsection “Iron mineralization and storage by the IMEF-system”:

" Thus, IMEF-systems are able to store substantially more iron than any known ferritin system (2,000 – 4,000 Fe atoms) (Andrews, 1998; Harrison and Arosio, 1996).”

The following passages discussing our findings and comparing them with other known systems were already present in our initial submission:

Subsection “Overall structure of the cargo-loaded IMEF encapsulin”:

“E-loops are located at capsomer interfaces and their relative orientation plays a key role in determining the overall topology and triangulation number of encapsulin compartments as evidenced by comparison of the IMEF T = 4 monomer with T = 1 (Thermotoga maritima), T = 3 (Pyrococcus furiosus) and T = 7 (HK97 phage) capsid proteins (Figure 1C).”

Subsection “Pores in the IMEF encapsulin shell”:

“This is similar to the negatively charged pores in ferritin systems that guide positively charged iron to the ferritin interior (Arosio et al., 2017). In no other encapsulin system are all pores negatively charged indicating that pores in the IMEF-system are optimized for attracting and channeling positively charged ions. The 2-fold pores observed at the interface of two capsomers in T = 1 and T = 3 encapsulins are not present in the IMEF-system (Nichols et al., 2017). The 3-fold pore forms the largest channel to the IMEF compartment interior and is 7.2 Å wide at its narrowest point, substantially larger than previously reported encapsulin pores.”

Subsection “Pores in the IMEF encapsulin shell”:

“The 2-fold symmetry axes at the center of hexameric capsomers also represent potential channels, as observed in T = 3 systems (Nichols et al., 2017)”

Subsection “Non-covalent chainmail and thermal stability of the IMEF-system”:

“This architecture has only been observed in a number of viral capsids including the HK97 bacteriophage but not in a bacterial system. In contrast to HK97 where an isopeptide bond covalently links E-loops and P-domains (Duda, 1998), the IMEF encapsulin uses non-covalent interactions.”

Subsection “Structure and analysis of the IMEF cargo protein”:

“This IMEF ferroxidase motif differs from known examples and represents an alternative way of forming an inter-subunit ferroxidase center (Figure 3D).”

Subsection “TP-mediated cargo-shell co-assembly”:

“The main TP binding sites surrounding the 2-fold symmetry axes are formed by conserved residues of the P-domain and N-terminal helix (Figure 2—figure supplement 2) similar to the T. maritima T = 1 encapsulin system (Sutter et al., 2008). No TP binding site has been identified for T = 3 encapsulins yet.”

Subsection “Iron mineralization and storage by the IMEF-system”:

“Selected area electron diffraction (SAED) further indicates that this mineralized material is amorphous (Figure 4—figure supplement 1B,C), similar to bacterioferritin systems (Andrews et al., 1993). The high P content and amorphous cores described for the IMEF encapsulin are similar to bacterioferritin systems (Aitken-Rogers et al., 2004; Mann, Bannister and Williams, 1986). It has been hypothesized that amorphous material can be more readily mobilized under iron-limited condition than crystallized iron mineral (Watt, Frankel, Jacobs, Huang and Papaefthymiou, 1992; Watt, Hilton and Graff, 2010).”

Subsection “Iron mineralization and storage by the IMEF-system”:

“In contrast to ferritin systems, IMEF encapsulins are two-component systems with the catalytic activity separated from the protein shell. The IMEF cargo protein is flexibly tethered and primarily localizes 4.5 nm away from the capsid interior.”

2) The authors state that 'A newly discovered class of protein organelles called encapsulin nanocompartments are implicated in microbial iron and redox metabolism and have so far only been shown to be involved in oxidative stress response (Giessen and Silver, 2017; He et al., 2016; McHugh et al., 2014; Sutter et al., 2008).', but cite a paper (McHugh, 2014) in which iron storage by an encapsulin is well documented. The authors' own work (Giessen and Silver, 2017) has previously shown that IMEF-Enc mineralises iron in vivo. I would therefore consider it established that encapsulins can function in iron storage.

We agree and have adjusted the wording in the Introduction accordingly:

“A newly discovered class of protein organelles called encapsulin nanocompartments have been shown to be involved in microbial iron storage and redox metabolism (Giessen and Silver, 2017; He et al., 2016; McHugh et al., 2014; Sutter et al., 2008).”

3) Other issues related to clarity:- Is IMEF a system or is IMEF a cargo protein?

We agree that this can be confusing given the fact that encapsulin systems are often named after their cargo proteins (e.g. peroxidase systems etc.). To clarify this, we have made sure that we always refer to the “IMEF cargo protein” when talking about the actual cargo protein and the “IMEF-system” when referring to the newly characterized overall iron mineralization system.

-Retention of the 'cargo protein' name instead of a protein name based on homology and presumptive function allows questions to linger unnecessarily.

We agree and are now always referring the “IMEF cargo protein” instead of just the “cargo protein”.

- The main statement about what protein construct/assembly is produced for study is (subsection “Overall structure of the cargo-loaded IMEF encapsulin”) "we produced homogeneous IMEF cargo-loaded encapsulins". What does that mean? What proteins were expressed?

For clarification, we have added the following to the main text:

Subsection “Overall structure of the cargo-loaded IMEF encapsulin”:

“Using a recombinant system for the expression of the two-gene IMEF operon containing the IMEF cargo protein gene and the encapsulin capsid protein gene, we produced homogeneous IMEF cargo-loaded encapsulins (Figure 1—figure supplement 1B).”

- Subsection “Overall structure of the cargo-loaded IMEF encapsulin”: "as evidenced by comparison of the IMEF T =4 monomer with T = 1, T = 3 and T = 7 capsid proteins." What capsid proteins? Is this referring to all encapsulin and HK97 proteins or something else?

We have added species information to the main text to clarify which capsid proteins we are referring to.

- There are places where "the" should probably be "a" instead, where a new idea hasn't been introduced previously. [subsection “Overall structure of the cargo-loaded IMEF encapsulin” on the flexibility of a linker in the cargo protein].

We have incorporated the suggested change.

- Subsection “Structure and analysis of the IMEF cargo protein” says that a phylogeny analysis shows IMEF is a member of the Flp superfamily, but could not be detected as the sequence level. What is meant here? That the IMEF protein has sequence similarity to other proteins whose structures were known and could be assigned to the Flp superfamily despite not being able to detect sequence similarity to other Flp members?

Low sequence similarity of the IMEF cargo protein to members of the Flp superfamily could indeed be detected via sequence alignments. However, it was not possible to detect any ferroxidase center-forming residues or motifs solely based on the sequence. A 3D structure was necessary to deduce which residues were involved in ferroxidase center formation which then allowed us to go back to the sequence and designate a new ferroxidase motif in the IMEF cargo protein. We have changed the text accordingly:

Subsection “Structure and analysis of the IMEF cargo protein”:

“Phylogenetic analysis revealed that the IMEF cargo protein is a member of the Flp superfamily and is most closely related to Dps proteins (Figure 3A and Supplementary file 3) but no known ferroxidase motifs could be detected based on the primary sequence alone (Andrews, 2010).”

A few technical issues also need to be addressed:4) The following points can be addressed changes to the text:- The issue of symmetry averaging and its presumptive effects on certain parts of the structure like the cargo are not handled cleanly (See subsection “Overall structure of the cargo-loaded IMEF encapsulin”; subsection “TP-mediated cargo-shell co-assembly”). The authors infer flexibility in some cases where lack of icosahedral symmetry in the presence of averaging would likely have the same effect. How would the cargo protein survive averaging if it sits as a single dimer bound to a pentamer at an icosahedral vertex?

We agree that this could have been handled more clearly and have changed the text accordingly:

Subsection “Overall structure of the cargo-loaded IMEF encapsulin”:

“No connection of cargo and shell density is visible, likely due to averaging or the flexibility of a 37 amino acid linker preceding the IMEF targeting peptide that directs and anchors the IMEF cargo to the shell interior. Averaging and linker flexibility likely also contribute to the lower resolution observed for the interior IMEF densities.”

- In subsection “Iron mineralization and storage by the IMEF system”, the logic about the shell permeability and kinetic curve shapes is unclear.

To address this point, we have expanded our explanation of the ferroxidase time curves as follows:

Subsection “Iron mineralization and storage by the IMEF-system”:

“However, assaying the IMEF cargo-loaded encapsulin results in a typical hyperbolic enzyme catalysis curve. These observations imply that the encapsulin shell controls the flux of iron to the inside of the compartment leading to a controlled and low concentration of soluble iron in the encapsulin interior. Therefore, the IMEF cargo protein is able to enzymatically oxidize the majority of ferrous iron before uncontrolled autocatalytic mineralization can lead to bulk precipitation of iron which would likely destroy the iron storage function of the IMEF-system (Figure 4 —figure supplement 4).”

- In subsection “Iron mineralization and storage by the IMEF system”, the idea of being "channeled to pores" is contrasted with diffusion in the next phrase. But presumptive pore transport here is presumably diffusive. The physical ideas need to be spelled out more carefully.

We agree that the wording was confusing and have changed it accordingly:

Subsection “Iron mineralization and storage by the IMEF-system”:

“This suggests that once iron enters the encapsulin interior via pores, it diffuses to the ferroxidase active site of the IMEF cargo, making it necessary to strictly control interior iron concentration to prevent runaway mineralization.”

- Grounds are lacking for the assertion in subsection “Overall structure of the cargo-loaded IMEF encapsulin” about the observed conformational diversity being important for pore function.

We agree and have changed the text accordingly:

Subsection “Overall structure of the cargo-loaded IMEF encapsulin”:

“A-domain loops form compartment pores and are likely adapted to optimize the particular function of a given encapsulin, for example ROS detoxification or iron mineralization. In addition, local resolution maps indicate that E-loops and A-domain loops represent the most flexible parts of the shell which suggests a certain structural flexibility of the pores formed by A-domain loops (Figure 1—figure supplement 4).”

- In subsection “Non-covalent chainmail and thermal stability of the IMEF system”, the absence of a patent pore is not evidence for a gated pore.

We agree and have changed the wording:

Subsection “Pores in the IMEF encapsulin shell”:

“This observation combined with the flexibility observed for loops around the 2- and 5-fold symmetry axes in local resolution maps (Figure 1—figure supplement 4) could indicate the presence of gated pores in encapsulins that may regulate ion flux to the compartment interior, similar to some ferritins (Theil, Liu and Tosha, 2008).”

- More caution is required on the claim of ions and density in the central regions of the capsid oligomers. For one, averaging often accentuates noise on symmetry axes. But further the identities/charge of any molecules there are entirely unknown; the densities could be water for example.

We agree and have changed the text to:

Subsection “Pores in the IMEF encapsulin shell”:

“Extra cryo-EM density is observed at the center of both the 3-fold and 5-fold pores. This could be a result of averaging accentuating noise on symmetry axes or potentially represent bound ions (e.g. Fe^2+/3+^) or even water molecules.”

5) The authors discuss the probability that features of the cryo-EM map, including the IMEF densities, are artifacts of averaging, which is almost certainly the case. This could be mitigated by symmetry expansion (relion_particle_symmetry_expand) and focussed classification/refinement for a clearer picture of the IMEF protein within the encapsulin shell. This is not essential, but would strengthen the paper considerably.

We thank the reviewers for this excellent suggestion. We have carried out the suggested analysis and have included an additional figure supplement (Figure 1—figure supplement 3) as well as a novel paragraph in the main text and Materials and methods section:

Subsection “Overall structure of the cargo-loaded IMEF encapsulin”:

“To further investigate and better resolve the cargo densities, we applied an approach combining symmetry expansion and focused classification with residual signal subtraction (Figure 1—figure supplement 3). This approach was able to separate cargo densities bound at slightly different locations indicating that the symmetry observed for the cargo densities (Figure 1b) is a result of averaging. The observed non-symmetrical densities are still weak compared to the shell density. At low threshold values possible connections between cargo densities and the shell are visible, potentially representing the linker connecting the cargo with the bound TP (Figure 1—figure supplement 3).”

Subsection “Symmetry expansion and focused classification”:

“In an attempt to better resolve cargo density within the encapsulin shell we used an approach combining symmetry expansion and focused classification with residual signal subtraction. Prior to symmetry expansion and focused classification, particles were binned to a box size of 192 with a corresponding pixel size of 3.41Å. Following refinement of binned particles with icosahedral symmetry, a 60Å low-pass filtered mask of a hexameric encapsulin shell unit with associated cargo density was generated (Figure 1—figure supplement 3A). Symmetry expansion was performed with relion_particle_symmetry_expand specifying “I” symmetry to generate a new particle stack with 60x increased particle number. Residual signal subtraction was performed as described previously (Bai, Rajendra, Yang, Shi and Scheres, 2015) to subtract encapsulin shell and cargo densities outside of the 60Å low-pass filtered mask from the symmetry expanded particle dataset (Figure 1—figure supplement 3B). Focused classification without alignment and without applied symmetry was then performed in Relion3.0 to resolve cargo density bound in different configurations to the encapsulin shell and potential connections between the cargo and targeting peptide (Figure 1—figure supplement 3C).”